



# Air quality impacts of COVID-19 lockdown measures detected from space using high spatial resolution observations of multiple trace gases from Sentinel-5P/TROPOMI

Pieternel F. Levelt[1,2], Deborah C. Stein Zweers[1], Ilse Aben[3], Maite Bauwens[4], Tobias Borsdorff[3], Isabelle De Smedt[4], Henk J. Eskes[1], Christophe Lerot[4], Diego G. Loyola[5], Fabian Romahn[5], Trissevgeni Stavrakou[4], Nicolas Theys[4], Michel Van Roozendael[4], J. Pepijn Veefkind[1,2], Tijl Verhoelst[4]

[1]Royal Netherlands Meteorological Institute (KNMI), De Bilt, 3731GA, The Netherlands
[2]University of Technology Delft (TU Delft), Delft, 2628 CN, the Netherlands
[3]Netherlands Institute for Space Research (SRON), Utrecht, 3584 CA, The Netherlands
[4]Royal Belgian Institute for Space Aeronomy (BIRA-IASB), Brussels, 1180, Belgium
[5]German Aerospace Centre (DLR), Oberpfaffenhofen, Wessling, 82234, Germany

*Correspondence to*: Deborah C. Stein Zweers (stein@knmi.nl)

**Abstract.** The aim of this paper is two-fold: to provide guidance on how to best interpret TROPOMI trace gas retrievals and to highlight how TROPOMI trace gas data can be used to understand event-based impacts on air quality from regional to city-scales around the globe. For this study, we present the observed changes in the atmospheric column amounts of five trace gases ($NO_2$, $SO_2$, CO, HCHO and CHOCHO) detected by the Sentinel-5P TROPOMI instrument, driven by reductions of anthropogenic emissions due to COVID-19 lockdown measures in 2020. We report clear COVID-19-related decreases in $NO_2$ concentrations on all continents. For megacities, reductions in column amounts of tropospheric $NO_2$ range between 14% and 63%. For China and India supported by $NO_2$ observations, where the primary source of anthropogenic $SO_2$ is coal-fired power generation, we were able to detect sector-specific emission changes using the $SO_2$ data. For HCHO and CHOCHO, we consistently observe anthropogenic changes in two-week averaged column amounts over China and India during the early phases of the lockdown periods. That these variations over such a short time scale are detectable from space, is due to the high resolution and improved sensitivity of the TROPOMI instrument. For CO, we observe a small reduction over China which is in concert with the other trace gas reductions observed during lockdown, however large, interannual differences prevent firm conclusions from being drawn. The joint analysis of COVID-19 lockdown-driven reductions in satellite observed trace gas column amounts, using the latest operational and scientific retrieval techniques for five species concomitantly is unprecedented. However, the meteorologically and seasonally driven variability of the five trace gases does not allow for drawing fully quantitative conclusions on the reduction of anthropogenic emissions based on TROPOMI observations alone. We anticipate that in future, the combined use of inverse modelling techniques with the high spatial resolution data from S5P/TROPOMI for all observed trace gases presented here, will yield a significantly improved sector-specific, space-based analysis of the impact of COVID-19 lockdown measures as compared to other existing satellite observations. Such analyses will further enhance the scientific impact and societal relevance of the TROPOMI mission.



**Key words:** Air quality, Trace gases, Sentinel-5P, TROPOMI, COVID-19, emissions

## 1 Introduction

In an effort to limit the transmission of the SARS-CoV-2 virus responsible for the Coronavirus disease 2019 (hereafter referred as COVID-19), drastic lockdown measures were implemented around the globe in the first half of 2020. These policies led to dramatic reductions in human activity, especially in the transport and industrial sectors, resulting in large decreases in the concentration of air pollutants (Bauwens et al., 2020; Shi and Brasseur, 2020; Forster et al., 2020; Diamond and Wood, 2020; Kroll et al., 2020; Le Quéré et al., 2020; Guevara et al., 2021; Gkatzelis et al., 2021). These changes were observed over China as early as February 2020 (Bauwens et al., 2020; Liu et al., 2020; Zhang, Z. et al., 2020; Zhao, N. et al., 2020) and were detected later in many other countries as similar lockdown measures were adopted (Bauwens et al., 2020; Broomandi et al., 2020; Collivignarelli et al., 2020; Lee et al., 2020; Gkatzelis et al., 2021).

The TROPOspheric Monitoring Instrument (TROPOMI; Veefkind et al., 2012; Ludewig et al., 2020) on board the European Copernicus Sentinel-5 Precursor (S5P) satellite, launched on 13 October 2017, is specifically designed for tropospheric monitoring on the global scale and has a daily revisit time. Compared to its predecessor OMI, TROPOMI's highest spatial resolution ($3.5 \times 5.5$ km$^2$) is about 16 times better and its signal-to-noise ratio per ground pixel is substantially higher. This results in a spectacular improvement in measurement sensitivity for relevant air quality products, including $NO_2$, $SO_2$, HCHO, and CHOCHO, thus enabling the study of rapid emission changes for even smaller sources as compared to previous instruments. For CO measurements, the daily global coverage of TROPOMI at a resolution of $7 \times 5.5$ km$^2$ represents a huge improvement to its predecessor SCIAMACHY (Bovensmann et al., 1999; Borsdorff et al., 2016; Borsdorff et al., 2017) with a spatial resolution of $120 \times 30$ km$^2$.

The observations from TROPOMI thus provide a unique opportunity to observe the magnitude and timing of the changes in tropospheric trace gas constituents, resulting from unprecedented COVID-19 lockdown measures. The initial TROPOMI observations of dramatic reductions in $NO_2$ concentrations over regions with strictly enforced lockdowns, over China in particular, triggered a high level of interest worldwide, and initiated a large number of studies, mainly aimed at regional scales and largely focused on $NO_2$. However, the unparalleled capacity of TROPOMI to provide relevant information on COVID-19 driven emission reductions based on multiple species measurements has not been exploited yet. The objective of this work is to investigate the COVID-19 driven changes in the concentration of five trace gases ($NO_2$, $SO_2$, CO, HCHO, and CHOCHO) from the global level down to individual cities using state-of-the-art TROPOMI operational and scientific data products. More specifically, we aim to

1. Expand the analysis of tropospheric $NO_2$ to all continents.

A large body of studies investigated the impact of the COVID-19 lockdowns on $NO_2$ concentrations (e.g. Bauwens et al., 2020; Baldasano, 2020; Huang et al., 2020), at regional and continental scale. Here, we analyze the time series of $NO_2$ measurements from a single satellite instrument for globally distributed locations on regional to city scales. In doing so, we further demonstrate the unique capabilities of how the TROPOMI instrument can be used to consistently track changes in air quality and anthropogenic emissions across the globe.



2.    Explore the high spatial resolution and simultaneous TROPOMI observations of $NO_2$, $SO_2$, CO, HCHO, and

74        CHOCHO.

While all of these gases have significant anthropogenic sources, they differ in their relative contribution to the energy,
industry, and transport sector emissions, and each sector exhibits a different response to COVID-19 lockdown
measures. Therefore, the combination of several TROPOMI trace gas products contains additional information on
sector-specific emissions and COVID-19 lockdown-induced changes in atmospheric composition. We show that
meaningful trends and source detection can be obtained by using the unprecedentedly high spatial resolution of
TROPOMI data and by averaging that data over relatively short time periods. Although this is in large part the result
of the improved sensitivity of the instrument, we also introduce new developments in trace gas retrieval techniques
and ad-hoc corrections to enhance the sensitivity of the TROPOMI datasets to even smaller emissions and smaller
changes in emissions. In order to achieve these goals, we discuss the strengths and limitations of each of the retrievals
for tracking global to city-scale changes.
In the next section, the TROPOMI data will first be described in general terms, followed by a description per species
to address the retrieval methods, as well as a description of how we handle each data product in this study. The goal
of this methods and data section is not only to explain how this study was conducted but also to provide guidance to
data users on how to best interpret and analyze TROPOMI trace gas data not only for lockdown-driven emission
changes but also for other event-driven changes. This will be followed by a section describing the impacts of COVID-
19 lockdown measures on all continents, using TROPOMI $NO_2$ data. The next two sections will describe the effect of
the lockdown measures on a regional scale by examining $NO_2$, $SO_2$, CO, HCHO, and CHOCHO for China and India.
The last section will feature an outlook of future applications for this type of analysis followed by conclusions.
**2    Methods and Data**
In this work, our analysis is primarily based on TROPOMI data for regional lockdown periods in 2020 as compared
to the same periods in 2019 and will be presented in the broader context of the TROPOMI operational data record,
which started on 30 April 2018. We make use of observations from the TROPOMI instrument on board S5P which is
a push-broom imaging spectrometer (Veefkind et al., 2012) measuring in the ultraviolet (UV), visible (VIS), near-
infrared (NIR), and shortwave infrared (SWIR) spectral bands selected to cover absorption regions for clouds and a
large number of trace atmospheric constituents. Using the spectral radiance measurements from TROPOMI,
atmospheric concentrations of different gases are retrieved as well as cloud and aerosol properties. For this work, we
use the following TROPOMI data products: $NO_2$, $SO_2$, CO, HCHO and CHOCHO as summarized in Table 1. We did
not include the following TROPOMI data products: tropospheric ozone columns, due to the tropics-only spatial
coverage; methane, due to an even longer atmospheric lifetime than CO where its sources were not as impacted by
lockdown measures; and aerosol index, designed to highlight long-range transported and/or elevated plumes of smoke,
dust, and/or ash and which is not a quantitative measure of aerosol amount nor sensitive to near-surface emissions.
The S5P satellite flies in a Sun-synchronous orbit, with a local overpass time of 13:30. TROPOMI has a 2600 km
wide swath, providing near-daily global coverage. The spatial sampling of TROPOMI varies over the spectral bands.
The nadir sampling at the start of the operational period on 30 April 2018 was approximately 3.5 x 7 $km^2$ (across- x



along-track) for the ultraviolet and visible bands, and 7 x 7 km$^2$ in the shortwave infrared band. On 6 August 2019,
after implementation of a modified co-adding scheme, the sampling for these bands was improved to 3.5 x 5.5 km$^2$
and 7 x 5.5 km$^2$, respectively.
TROPOMI observations are being widely used within and beyond the scientific community and so it is crucial to
provide information on how these observations can best be used, interpreted, and analyzed. The COVID-19 lockdown
periods provide a unique use-case for the TROPOMI lead algorithm developers to highlight important differences in
the individual atmospheric lifetime and detectability of each trace gas and show how these characteristics are key to
the interpretation of the concomitant observations. It is not sufficient, for example, to illustrate lockdown-driven
changes in emissions simply by selecting a single day or week of TROPOMI column data for a given region as
measured during a lockdown period to the same day or week from year(s) prior (Braaten et al., 2020). We go further
to address the importance of delineating meteorological and seasonal variability from lockdown-driven changes in
emissions.
Therefore, we start this methods and data section with a general overview of considerations for the data user to take
into account for analyses aimed at the quantification of changes in the emission of these trace gases. Next, in dedicated
subsections, we provide a summary of the most relevant documentation and retrieval methods employed for each trace
gas (see Table A1). Even though each retrieval is based on the analysis of the amount of trace gas specific absorption
in measured radiance spectra, methods differ significantly per species.

### 2.1   Understanding and Interpreting TROPOMI trace gas retrievals

For this paper we will focus on TROPOMI trace gas retrievals for $NO_2$, $SO_2$, CO, HCHO, and CHOCHO (See Table
1). To understand and interpret the TROPOMI measurements of these trace gas species and how they vary with respect
to COVID-19 lockdown measures, it is necessary to consider their sources, variability through the atmospheric
column, and their atmospheric lifetimes. Although the mechanisms for the emission of each gas are different, there
are several common anthropogenic emission sources, most notably from transportation and industry, as listed in Table
1 which were significantly impacted by lockdown measures.

**Table 1: Summary of the retrieval spectral range, atmospheric lifetime, and primary emission sources, for each trace gas**
**addressed in this study.**

| Trace Gas (retrieval reference) | Spectral Range | Typical lifetime | Primary emission sources |
|---|---|---|---|
| $NO_2$ (van Geffen et al., 2019) | 405-465 nm | 2 to 12 hours | - Transportation<br>- Industry<br>- Power generation<br>- Biomass burning |
| $SO_2$ (Theys et al., 2021) | 310.5-326 nm | 6 hours to several days | - Power generation<br>- Industry<br>- Transportation<br>- Volcanoes[1] |





| CO (Landgraf et al., 2016) | 2324–2338 nm | Weeks to a month | - Power generation<br>- Industry<br>- Transportation<br>- Residential cooking and heating<br>- Biomass burning<br>- Oxidation of biogenic hydrocarbons<br>- Methane Oxidation |
|---|---|---|---|
| HCHO (De Smedt et al., 2018) | 328.5-359 nm | Several hours<br><br>(lifetime of NMVOC precursors up to several days) | Primary and secondary product (NMVOC precursors) from:<br>- Biogenic emissions<br>- Biomass burning<br>- Industry<br>- Transportation |
| CHOCHO (Lerot et al., 2010, 2020) | 435-460 nm | 2 to 3 hours | Primary and secondary product (NMVOC precursors) from:<br>- Biogenic emissions<br>- Biomass burning<br>- Transportation<br>- Industry |

[1]Volcanic emissions are not significant for this work.

A brief evaluation of how the sources of these trace gases were or were not affected by lockdown-driven changes
lends insight into expected changes. In general, primary production trace gases, like $NO_2$ and $SO_2$ with relatively short
atmospheric lifetimes exhibit emission changes most clearly and rapidly. Although $NO_2$ and $SO_2$ are both important
primary production anthropogenic pollutants, their sectoral sources are different. For instance, the impact of lockdown
on the transportation sector is expected to have a bigger impact on $NO_2$ than $SO_2$, since this sector is responsible for
about 30% of the global NOx emissions and only 1% of the global $SO_2$ emissions, according to the CAMS-ANT
inventory (Granier et al., 2019). On the other hand, $SO_2$ emissions are more likely to be impacted by possible changes
in power generation, since this sector accounts for 52% of the global $SO_2$ emissions and only 30% of the global NOx
emission (Granier et al., 2019).
For CO, secondary production by methane oxidation and the oxidation of (biogenic) hydrocarbons accounts for at
least 60% of the total atmospheric CO, followed by contributions from biomass burning and fossil fuel use (Müller et
al., 2018; Holloway et al., 2000). Anthropogenic CO emissions originate from the industry, transportation, and
residential sectors and account for about 30% of the global emissions (Granier et al., 2019). Although local impacts
of lockdown are likely for locations with strong anthropogenic CO emissions, overall a much smaller lockdown-driven
impact is expected for CO based on its longer atmospheric lifetime and smaller contributions from lockdown affected
sources.
Both HCHO and CHOCHO are short-lived indicators of non-methane volatile organic compound (NMVOC)
emissions resulting from biogenic processes, large biomass burning events, and anthropogenic activities (Millet et al.,



2008; Fu et al., 2008; Stavrakou et al., 2009; Bauwens et al., 2016; Chan Miller et al., 2016). They are mostly produced
as secondary products from oxidation of other NMVOCs but are also directly emitted from combustion and industrial
processes, although to a lesser extent. In general, the relative production of CHOCHO from such combustion processes
and from the oxidation of aromatics, originating mostly from the industrial sector, is higher than for HCHO. Thus, the
CHOCHO response to changes in anthropogenic emissions is expected to be stronger (Chan Miller et al., 2016; Cao
et al., 2018).
It is important to note that the retrievals provide information on the tropospheric or total column amount of these
gases, because the spectra contain limited information on their vertical distribution in the atmosphere. TROPOMI
observations thus provide a two-dimensional representation of the three-dimensional atmosphere. The vertical profiles
of each trace gas vary significantly depending on the injection height of the emissions and atmospheric lifetime (see
Table 1). For example, NOx emissions at the surface result in $NO_2$ vertical profiles that peak in the near-surface layer
(lowest 1-2 km of the troposphere), due to the short lifetime of $NO_2$. Similarly, $SO_2$ has a vertical profile which
generally peaks in the lower troposphere. CO on the other hand, has a lifetime of weeks to a month (depending on the
reaction with the hydroxyl radical) and can be transported over great distances, both horizontally and vertically.
Therefore, CO even though it is often co-emitted with $NO_2$, has a significantly higher background concentration
throughout the troposphere as compared to $NO_2$. HCHO and CHOCHO have lifetimes of a few hours but are generally
formed in the atmosphere via secondary production processes, which leads to an intermediate profile shape as
compared to $NO_2$ and CO.
In addition to vertical profiles that vary per trace gas species, the vertical sensitivity of the TROPOMI measurements
also varies per species. For the trace gases retrieved in the UV and VIS ranges, the sensitivity decreases towards the
surface so that the accuracy of the retrieved column depends on a well-characterized a priori knowledge of the vertical
distribution. Due to scattering, the near-surface sensitivity is lower in the UV ($SO_2$, HCHO) than in the VIS ($NO_2$ and
CHOCHO). In the SWIR range, the vertical sensitivity is more constant. As part of the retrieval process, a priori
vertical profiles of each trace gas are scaled to match the measured tropospheric column. An uncertainty in the
retrieved column amount or vertical column density (VCD) is associated with inherent differences between the true
and a priori vertical profiles. However, the averaging kernels, which are reported in the data products, can be used to
replace the a priori profiles with custom profiles (e.g. Eskes and Boersma, 2003; Eskes et al., 2020) thereby reducing
the corresponding uncertainty. In this study, we mostly focus on relative changes in VCDs and use standard a priori
profiles for each data product. Therefore, the uncertainty related to the vertical profile is rather small (as detailed in
Sect. 2.2 through 2.6). Another contribution to this error is the use of partly cloudy scenes by each retrieval which
increases the amount of data available but does change the vertical sensitivity. The cloud fraction threshold for each
trace gas is described in Sect. 2.2 through 2.6. In future studies, the averaging kernels could be used for inversion
modelling of emissions thus eliminating this error completely.
TROPOMI observes atmospheric concentrations of trace gases averaged over a vertical column, which is not the
same as a direct measurement of the (near-surface) emission. The column averaged amount of a given trace gas
measured at a certain location depends not only on emission and deposition, but also on atmospheric transport and
(photo)chemical reactions. Note that the background concentration is higher for trace gases with a longer atmospheric





lifetime. In turn, enhanced background concentrations will increase the relative importance of atmospheric transport
versus local emissions. Local $NO_2$ emissions have a relatively large impact on the measured column amounts, while
for CO the contribution of remote sources can in some cases be superimposed on local emissions thus making the
interpretation more difficult. To attribute a change in concentration to a corresponding change in local emissions, the
effects of meteorology and chemical lifetime must be accounted for as well.
While emissions can be estimated from satellite observations using data-driven methods (Beirle et al., 2019, Beirle
et al., 2021; Fioletov et al., 2016; Goldberg et al., 2019) or using complex inverse modelling techniques (e.g. Millet
et al., 2008; Stavrakou et al., 2009; Bauwens et al., 2016; Ding et al., 2020; Miyazaki et al., 2020; Borsdorff et al.,
2019; Borsdorff et al., 2020), here we use a more qualitative approach to probe emission changes. First we compare
the concentrations in 2020 with those from the same period from earlier years and then carry out additional analysis
to separate the lockdown-driven variability from seasonal and meteorological variability taking in account emission
changes driven by mechanisms.
**2.2     Nitrogen dioxide ($NO_2$)**
The tropospheric column of nitrogen dioxide ($NO_2$) is a TROPOMI operational data product (Veefkind et al., 2012;
doi.org/10.5270/S5P-s4ljg54). Product versions are listed in the Product Readme File (PRF, Eskes and Eichmann,
2019a). The retrieval method is described in detail in the $NO_2$ Algorithm Theoretical Basis Document (ATBD, van
Geffen et al., 2019). The data product and data usage are described in in the $NO_2$ Product User Manual (PUM, Eskes
et al., 2020). The dataset used for most of $NO_2$ analyses cover the period from 1 January 2018 to 30 May 2020. For
Europe, the dataset was extended through 31 August 2020.
The retrieval algorithm derives $NO_2$ information from spectral range 405-465 nm and is largely based on the OMI
$NO_2$ retrieval developments implemented during the EU QA4ECV project (Boersma et al., 2018). The retrieval
consists of three steps. The first step is based on the DOAS approach, in which the total slant column of $NO_2$ is
retrieved from the TROPOMI spectra, as discussed in van Geffen et al. (2020). The second step is the estimation of
the 3-D stratospheric distribution of $NO_2$ based on an assimilation of the TROPOMI slant column data of previous
days using the chemistry-transport model TM5-MP (Williams et al., 2017) run at 1° x 1°. This assimilation is set up
to predominantly make use of measurements over clean areas (e.g. ocean and remote land regions) with limited
tropospheric $NO_2$. The third step is the conversion of the tropospheric slant column (total minus stratosphere) into a
tropospheric vertical column by combining radiative transfer calculations with a priori profile shapes from the TM5-
MP model. The data product is very comprehensive and provides all the input (such as surface and cloud information)
and intermediate products.
The tropospheric column is delivered with corresponding averaging kernels and a detailed error estimate. The
random error on the slant column is discussed in van Geffen et al. (2020), and is on the order of $0.56 \times 10^{15}$ molec cm$^{-}$
$^{2}$ for individual measurements after 6 August 2019 (for pixel size 3.5 x 5.5 km$^2$). This translates to only small random
errors in the total columns on the order of $0.2 \times 10^{15}$ molec cm$^{-2}$. Uncertainties in the estimate of the local stratospheric
column amount is of the same order of magnitude. The uncertainty related to the computation of the air mass factor
(AMF) is much more significant for tropospheric columns over polluted areas. The AMF uncertainties are driven by





the treatment of surface albedo, clouds, aerosols, and profile shape. Such errors are multiplicative, and are of the order
of 20-60% depending on the geographical location, time of day, and season. These uncertainties are modelled for
individual observations and are provided in the data product.
As for all operational TROPOMI data products, a quality assurance value (qa_value) is provided to filter the data
and remove lower quality data where, the recommended threshold value depends on the application. For direct
visualization or gridding applications a qa_value greater than 0.75 is recommended. For comparisons with models and
data assimilation through the use of the averaging kernels, a relaxed qa_value of greater than 0.5 may be used. In this
study we use $NO_2$ retrievals with a qa_value greater than 0.75. Application of this qa_value threshold corresponds to
data with mostly clear-sky conditions (cloud radiance fractions < 0.5) and implies that the data is filtered to remove
retrievals which do not meet certain quality criteria as described van Geffen et al. (2019).
Several recent papers discuss the validation of the $NO_2$ product against independent observations (Verhoelst et al.,
2021; Tack et al., 2021; Judd et al., 2020; Dimitropoulou et al., 2020; Ialongo et al., 2020). The main findings can be
summarized as follows: the stratospheric and slant columns are in good overall agreement with other satellite
measurements (van Geffen et al., 2020) and with ground-based observations (Verhoelst et al., 2021). However, the
tropospheric column presents a negative bias of the order of 30% with respect to ground-based remote sensing
reference observations (Verhoelst et al., 2021; Dimitropoulou et al., 2020), as well as with imaging data from airborne
measurements (Judd et al., 2020; Tack et al., 2021). Although the origin of this bias remains unclear and may be due
to several causes, validation results indicate that it scales linearly with the retrieved tropospheric column amount
(Verhoelst et al., 2021; see Fig. C1). As a result, (COVID-related) relative changes in the $NO_2$ column, e.g., (2020-
2019)/2019, should be largely insensitive to this bias.

### 2.3   Sulphur dioxide (SO₂)

Initial analyses were performed using the TROPOMI operational data product for $SO_2$ (Theys et al., 2017). However,
biases present in those data (Fioletov et al., 2020) hamper the detection of the type of small changes in $SO_2$, typically
on the order of -0.1 DU, that are under investigation in this work. Therefore, an alternative retrieval scheme was
applied, the so-called COvariance-Based Retrieval Algorithm (COBRA; Theys et al., 2021). In brief, the approach
considers a set of $SO_2$-free spectra in the wavelength range 310.5-326.0 nm (from TROPOMI band 3) to represent the
radiance background variability, in the form of a covariance matrix. The latter is updated for each orbit, TROPOMI
row, and per latitude band. The covariance matrix is used to determine the $SO_2$ slant columns from individual spectral
measurements using an optimally weighted single parameter retrieval (see Walker et al., 2011). We note that COBRA
does not recalculate air mass factors (AMF). These are simply extracted from the operational product to convert $SO_2$
slant columns into vertical columns (VCDs). Compared to the operational DOAS results, COBRA significantly
improves the $SO_2$ VCDs, both in terms of precision and accuracy. Because the approach empirically accounts for all
sources of systematic variability in the measured signal, large-scale biases typically observed with the DOAS approach
are efficiently removed leading to a large gain in sensitivity (see Fig. C2).
In this study, we use $SO_2$ retrievals under clear-sky conditions (cloud fractions less than 30%) with solar zenith
angles lower than 60°, and we eliminate 25 swath edge pixels from each side of the orbit swath (450 pixels wide). The





random error in the $SO_2$ vertical columns is rather small in the range of 0.5-1.0 DU, and can be largely reduced by
data averaging. Errors due to spectral interferences are estimated to be very low, about 0.05 DU. Remaining systematic
uncertainties are mostly from the auxiliary data used in the AMF calculation, and are in the 30-50% range. The dataset
used for this analysis covers the period from May 2018 to June 2020.
**2.4    Carbon monoxide (CO)**
The total column of carbon monoxide (CO) is a TROPOMI operational data product obtained using TROPOMI 2.3
micron measurements (Veefkind et al., 2012; doi.org/10.5270/S5P-1hkp7rp). Product versions are listed in the Product
Readme File (Landgraf et al., 2020). The data product and data usage are described in in the CO Product User Manual
(Apituley et al., 2018). This CO retrieval uses the Shortwave Infrared CO retrieval (SICOR) algorithm method and is
described in detail in the CO Algorithm Theoretical Basis Document (Landgraf et al., 2018). The algorithm software
is based on a scattering forward model and retrieves trace gas columns simultaneously with effective cloud parameters
(cloud height, cloud optical thickness) from the SWIR channel to account for cloud contaminated measurements
(Landgraf et al., 2016, 2018). The inversion deploys a profile scaling approach by which a vertical CO reference
profile is scaled to obtain agreement between the forward simulation and the spectral measurement (Borsdorff et al.,
2014). The reference profile is based on a monthly averaged simulation from the global chemical transport model
TM5 and thus varies spatially and temporally (Krol et al., 2005). The vertical sensitivity of the retrieval for clear-sky
conditions is good throughout the atmosphere while measurements for cloudy conditions have reduced sensitivity
under the cloud (Borsdorff et al., 2018).
In this study, we use the CO retrieval for measurements under clear-sky and cloudy atmospheric conditions (cloud
altitude less than 5000m). This corresponds to filtering the dataset by using the quality assurance values (qa_value
greater than 0.5) that are supplied with the data product. CO retrievals under low cloud conditions perform well for
unpolluted scenes however can lead to e.g. lower CO values when pollution hot spots are present below the cloud due
to optical shielding and scattering (Borsdorff et al., 2018). Consequently, retrievals under cloudy conditions must be
considered with care, however they are essential to improve the data coverage especially over the oceans where clear-
sky measurements are hampered by the low reflectivity of water in the SWIR spectral range.
The CO retrieval skill lies well within the requirements of the TROPOMI mission (Veefkind et al., 2012) on accuracy
(< 15%) and precision (< 10%). This was shown by validation with ground-based FTIR measurements operated by
the Total Carbon Column Observing Network (TCCON). TROPOMI CO is biased high compared to TCCON by
about 6 ppb with a station to station variability of about 4 ppb (Borsdorff et al., 2018; Lambert et al., 2020). The
dataset used for this analysis covers the period from 1 January 2018 to 30 May 2020.
**2.5    Formaldehyde (HCHO)**
The tropospheric column of formaldehyde (HCHO) is a TROPOMI operational data product (Veefkind et al., 2012;
doi:10.5270/S5P-tjlxfd2). Product versions are listed in the HCHO Product Readme File (De Smedt et al., 2020a).
The data product and data usage are described in in the HCHO Product User Manual (PUM, Romahn et al., 2020).
The TROPOMI HCHO retrieval algorithm has been fully described in De Smedt et al. (2018) and in the HCHO ATBD





(De Smedt et al., 2020b). It is based on the DOAS method, and is directly inherited from the OMI QA4ECV product
(https://doi.org/10.18758/71021031). The fit of the slant columns is performed in the spectral interval of 328.5-359.0
nm. Reference spectra are updated daily using an average of Earth radiances selected in the Equatorial Pacific region.
The conversion from total slant to tropospheric vertical columns is performed using a look-up table of vertically
resolved air mass factors calculated at 340 nm. A priori vertical profiles are provided by the TM5-MP daily forecast
with a spatial resolution of 1 x 1 degree (Williams et al., 2017). Cloud properties are taken from the S5P operational
product Cloud as Reflecting Boundary (CRB; Loyola et al., 2018). In order to correct for any remaining offset and
striping due to instrumental artefacts or unknown misfits in the spectral retrieval, a background correction is applied
based on HCHO slant columns selected in the emission-free Pacific Ocean. The background HCHO vertical column,
due to the methane oxidation, is added using data from the TM5 model in the reference region. We use the quality
assurance values (qa_value greater than 0.5) to filter out observations presenting a solar zenith angle larger than 70°
or cloud fractions larger than 0.4.
The HCHO retrieval fulfils the requirements of the TROPOMI mission (Veefkind et al., 2012) on accuracy (40-
80%) and precision ($12 \times 10^{15}$ molec cm$^{-2}$). The precision of a single observation is estimated to be $5 \times 10^{15}$ molec cm$^{-2}$
in remote locations. The dispersion is naturally larger over polluted sites (from $7\text{-}10 \times 10^{15}$ molec cm$^{-2}$). Validation
using a global network of FTIR measurements indicates that TROPOMI HCHO columns present a negative bias over
high emission sites (-30% for HCHO columns larger than $7.5 \times 10^{15}$ molec cm$^{-2}$) and a positive bias for clean sites
(+20% for HCHO columns lower than $2.5 \times 10^{15}$ molec cm$^{-2}$) (Lambert et al., 2020; Vigouroux et al., 2020).
To characterize the HCHO interannual and seasonal variability, we have used the QA4ECV OMI dataset to construct
a climatology based on recent years (2010-2018). This is justified the good agreement between OMI and TROPOMI
HCHO columns which is better than 10% for most regions (Lambert et al., 2020). For our analysis, we use two-week
averaged columns. This reduces the random uncertainty to about 10%.
One of the main drivers of the observed HCHO variability is temperature, which has a direct impact on NMVOC
emissions and on the chemical production of HCHO (Stavrakou et al., 2018). It results in a strong correlation between
HCHO columns and surface temperatures. For this paper, we correct the HCHO concentrations for this meteorological
impact prior to using the data in the analyses. We introduce a temperature correction method (Zhu et al., 2017) based
on data from OMI for 2005-2020, and from TROPOMI for 2018-2020. In brief, this correction entails fitting a second-
order polynomial through daily HCHO columns reported as a function of the temperature. This novel analysis is
performed for each region and on the OMI and TROPOMI time series separately. On this basis, the temperature-
induced variations in HCHO are removed from the time series using local daily temperatures specified by ERA5-Land
2m meteorological datasets (Muñoz Sabater, 2019a; See Fig. C3). This correction is designed to minimize the impact
of temperature fluctuations on the HCHO anomalies. Finally, a polynomial obtained using a climatology of surface
temperatures is added to the differential HCHO columns, in order to reintroduce the natural seasonal cycle, assuming
the same temperature every year. These temperature-corrected HCHO columns are used throughout this paper. Note
that the difference with uncorrected HCHO columns is generally small (less than 10%), but can be significant when
looking for small effects such as those induced by COVID-19 related emission changes. The dataset used for this
analysis covers the period from May 2018 to June 2020.





**2.6    Glyoxal (CHOCHO)**
Glyoxal (CHOCHO) is not one of the TROPOMI operations data products. For this study we used the prototype data
product developed as part of the ESA S5p+I GLYRETRO project, which relies on scientific developments performed
using the GOME-2 and OMI instruments (Lerot et al., 2010). The algorithm is described in detail in the GLYRETRO
ATBD (Lerot et al., 2020). In brief, the retrieval approach consists of a DOAS-type spectral fit for the observed optical
depth with reference absorption cross-sections for glyoxal and other absorbing species ($NO_2$, $O_3$, $O_2$-$O_2$, liquid water
and water vapor, and the Ring effect) in the spectral interval of 435-460 nm to derive glyoxal slant column densities.
The latter are converted into tropospheric columns using calculated air mass factors, after application of a background
correction procedure aimed at reducing possible remaining (row-dependent) systematic biases. Air mass factors are
calculated following the formulation of Palmer et al. (2001), which combines box-air mass factors precomputed with
the radiative transfer model VLIDORT v2.7 (Spurr and Christi, 2019) and a priori glyoxal concentration profiles
provided by the MAGRITTE chemistry-transport model (Müller et al., 2018, 2019).
The glyoxal optical depth is very small ($< 5x10^{-4}$), which makes its retrieval very sensitive to instrumental noise and
to interferences with spectral signatures of species absorbing more significantly in the same spectral region. The first
factor introduces large random errors, in the range $6\text{-}10x10^{14}$ molec $cm^{-2}$, which can however be reduced by spatial-
temporal averaging, that is, using multiple observations averaged time and/or space. Systematic uncertainties are
dominated by spectral interferences, but also by uncertainties associated with the auxiliary data used as an input for
the AMF calculation. These uncertainties are estimated to be $2\text{-}3x10^{14}$ molec $cm^{-2}$ (~50% for source regions). To limit
uncertainties related to cloud contamination, glyoxal observations are only provided for scenes with effective cloud
fractions smaller than 20% (taken from the operational $NO_2$ product). As with HCHO, to account for seasonal and
interannual variability, a climatology of OMI CHOCHO columns was built to further delineate sources of variability
for glyoxal column amounts.
Validation of satellite glyoxal column observations is generally limited, mostly due to the scarcity of independent
ground-based data. However, a preliminary validation based on a few MAX-DOAS stations in Asia and Europe,
indicates that the satellite and ground instruments measure consistent glyoxal tropospheric column amounts with mean
differences generally less than $2x10^{14}$ molec $cm^{-2}$, except in particular conditions such as low sun elevation or for
stations that are frequently covered by clouds (Alvarado et al., 2020). The dataset used for this analysis covers the
period from May 2018 to June 2020.
**3    Global Observations of Nitrogen Dioxide**
TROPOMI measurements of tropospheric $NO_2$ column amount are well-suited for detecting emission from a variety
of anthropogenic sources including traffic, power plants, and industry. The atmospheric lifetime of $NO_2$ and its vertical
profile shape dictate that the high spatial resolution measurements from TROPOMI can readily capture rapid week-
to-week changes in near-surface emissions from COVID-19 impacted cities and point sources. To give context and
overview, the global distribution of tropospheric $NO_2$ based on an annual average for 2019 with an oversampling
resolution of approximately 0.02° x 0.02° is illustrated in Figure 1. The high resolution of these measurements enables


further zooming to the regional, suburban, and city scale providing detailed information about spatial distributions. A
regional zoom-in over central South America reveals high $NO_2$ levels over the megacities of Rio de Janeiro, São Paulo,
Buenos Aires, and Santiago. A further zoom-in to central Chile and its capital Santiago is shown in Figure 1, focusing
on a shorter period from 23 March to 10 April 2020, which coincides with a region-specific COVID-19 lockdown
(Figure 2k), as compared to the mean tropospheric $NO_2$ column for March-April 2019. Note that the period in 2019
is chosen to be longer than 2020 in order to reduce the effects of natural variability, but the period is centered at the
beginning of April to avoid the influence of the seasonal $NO_2$ cycle. A strong reduction in the $NO_2$ tropospheric
concentration of about 40% is observed over Santiago during this period, and a 28% reduction is observed between
23 March and 15 May corresponding to the period when restrictions were eased (Figure 2k). Interestingly, a further
zoom shows that the relative reduction is not uniform over the city, reflecting differences in the mix of source
contributions for the different quarters of the city.




**Figure 1: Global distribution of NO₂ based on the annual average of tropospheric column amounts of NO₂ measured by TROPOMI for 2019 (top panel) shown in units of micromole per m². Using the same data, several zoom-in plots are shown in the middle and bottom panels: regional zoom-in for central South America (middle left) and a city-scale zoom-in over Santiago, Chile (middle right panels, comparing 23 March to 10 April 2020 with March-April 2019), over Paris (lower left, comparing 15 March to 15 April 2020 with March-April 2019) and over New Delhi (lower right, comparing 28 March to 22 April 2020 with April 2019). Note the different color scales in the three subpanels. The domain size of the panels is 1.5 x 1.0 degree for Paris, and 1.1 x 1.0 degree for New Delhi.**





Two more examples of lockdown-related $NO_2$ column reductions in major cities are shown for Paris and New Delhi
in Figure 1 with time windows selected to reflect region-specific lockdown periods. In Paris, the $NO_2$ levels for the
period 15 March to 15 April 2020 are about a factor of two lower than in March-April 2019 (see also Figure 4). For
New Delhi the reduction is even more striking in comparison to April 2019 (about a factor of 3, Figure 2c). Both Paris
and New Delhi also show significant reductions in background values around the cities. Background locations are
subject to a variety of wind directions and sometimes downwind of city plumes thus influencing background
concentrations. Such plumes are typically on the order of 100 km long, and, given the atmospheric residence time of
$NO_2$ (2-12 hours), these plumes can fill the small domains around Paris and New Delhi shown in Figure 1.

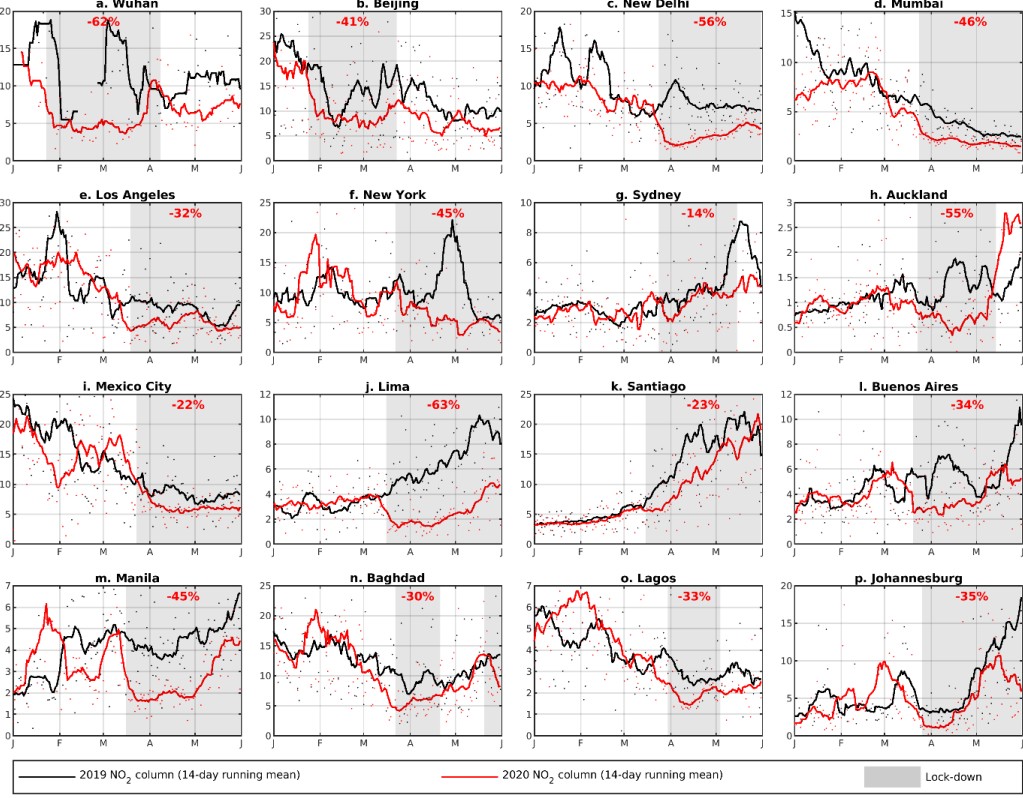


**Figure 2: Time series of TROPOMI $NO_2$ column amounts (in $10^{15}$ molec cm$^{-2}$) for selected cities for the period 1 January**
**to 1 June in 2019 (black dots) and 2020 (red dots). TROPOMI observations are averaged over a 25 x 25km$^2$ box around the**
**city center. The lines indicate the two-week running mean for 2019 (black) and 2020 (red). The grey zones indicate the**
**official lockdown period for each city. The reduction of the average $NO_2$ column during the lockdown period relative to the**
**same period in 2019 is given inset. Details about the lockdown dates are summarized in Table C2.**



The lockdown periods and the measures taken to mitigate the spread of the COVID-19 were rolled out on a country-
and often city-specific basis. Figure 2 illustrates the temporal evolution of $NO_2$ tropospheric columns from January to
May over large cities for different continents. The observed reductions in China and India are discussed in more detail
in Sect. 4 and 5. Detailed information about the lockdown measures adopted for those cities is given in Table C2. The
TROPOMI observations indicate substantial decreases in $NO_2$ during the lockdowns in all studied cities, but the
reductions vary significantly from one city to another.
In Wuhan, the first city to issue quarantines and lockdown measures, the observed $NO_2$ column drastically declined
(-60%) between 23 January and 8 April 2020 compared to the same period in 2019 (Table C2). This decrease is in
good agreement with estimated reductions for the period 11 February to 2 March 2020 based on TROPOMI $NO_2$ (-
43%, Bauwens et al., 2020) and in situ $NO_2$ observations in Wuhan (-55%, Shi and Brasseur, 2020). However, it
should be noted that there was strong day-to-day variability in the $NO_2$ column amount due to meteorological factors,
as well as missing data over Wuhan in February 2019 due to clouds. Model calculations by Liu et al. (2020) indicate
that meteorological variability could have led to increased $NO_2$ columns in 2020 compared to 2019, suggesting that
the observed $NO_2$ reductions underestimate the impact of emission reductions due to COVID-19. The partial lifting
of the restrictions on 8 April led to a progressive increase in $NO_2$ levels, yet remained lower than in 2019, likely
because the population was still advised to stay at home and schools remained closed. A similar response in $NO_2$
levels was observed in Beijing. The decreases were less pronounced (-40%) and are in excellent agreement with the
reported decrease based on in situ $NO_2$ measurements (-40%, Shi and Brasseur, 2020). The weaker response could be
due to the less drastic measures adopted in Beijing, because locally sustained COVID-19 cases were lower than in the
Hubei province (Leung et al., 2020). Strong $NO_2$ reductions were observed for other Chinese cities, like Nanjing,
Qingdao, and Zhengzhou, based on TROPOMI $NO_2$ observations (Bauwens et al., 2020).
India enforced strict restrictions of human activities on 24 March 2020 to tackle the spread of COVID-19. In New
Delhi and Mumbai, the onset of the lockdown induced a sharp decline in the observed $NO_2$ columns (by a factor of
2). The columns remained low during the entire lockdown period (-56% and -46%, respectively) (see Table 2 for
timing of lockdown phases). This is very much in line with the decreases reported in New Delhi based on $NO_2$ data
from monitoring stations, -53% (Mahato et al., 2020) and -48% (Jain and Sharma, 2020).
As compared to other cities, a very strong $NO_2$ decrease was observed in Lima (-63%), where strict regulations to
stay indoors were enforced (Collyns, 2020). A drastic drop in $NO_2$ compared to the 2019 levels marked the start of
the lockdown, and the levels remained very low throughout the entire lockdown period. The gradual increase of $NO_2$
columns in Lima and other Southern Hemispheric cities from January to May (Figure 2) reflects the natural seasonal
variation when levels peak during the Southern Hemispheric winter, as temperatures decrease and $NO_2$ lifetime
increases.
In Buenos Aires, the observed reduction was not as strong compared to Lima for the entire lockdown period (-34%,
Table C2), but was particularly marked during the first month of the lockdown (20 March through 20 April 2020),
due to a compulsory quarantine period and strict limitation of activities for many sectors. Although partial lifting of
measures was issued after 10 April for many provinces in Argentina, the measures in the Buenos Aires agglomeration
were maintained due to the elevated number of cases (Raszewski and Garrison, 2020). More moderate reductions are



found for Mexico City (-22%) and Santiago (-23%) during the lockdown in comparison to the same period in 2019,
that could be attributed to less strict adherence to and enforcement of lockdown measures (Uchoa, 2020; Pasley, 2020).
Strong reductions were observed over the entire lockdown period in the heavily hit cities in southwest Europe, Los
Angeles, and New York, with reductions ranging between -32% and -54% (Bauwens et al., 2020). It should be noted
however, that in these regions, the start of the lockdown period is generally less marked partly because the lockdowns
were not as strictly enforced in Europe and the U.S. as in China and India. Moreover, the observed TROPOMI data
displays a strong variability attributable to meteorology, e.g. over Paris, New York and Los Angeles in 2019.
In Sydney, the reduction was moderate (-14%) and delayed with respect to the onset of the measures (Figure 2).
This could be related to observations of less strict compliance in the early period of lockdown measures (New South
Wales Public Health, 2020). A rapid and strong decrease was observed for $NO_2$ column amount as a result of lockdown
measures in Auckland, New Zealand (-55%). Similarly, the lockdown measures in New Zealand were implemented
swiftly with high levels of compliance (Matthews, 2020). The end of the lockdown coincided with a strong increase
in $NO_2$ pollution, from $1.8 \times 10^{15}$ molec cm$^{-2}$ to $3 \times 10^{15}$ molec cm$^{-2}$ in the last three weeks of May.
In Africa, Nigeria is among the countries most affected by COVID-19 and reported the first confirmed case in sub-
Saharan Africa (Odunsi, 2020; Adigun and Anna, 2020). A two-week lockdown period was put in place for Lagos
starting 30 March. The $NO_2$ column amount decreased by 33% during the lockdown (Figure 2) with respect to the
same period of 2019 and remained lower even after the lifting of restrictions on 4 May (Table C2). An $NO_2$ column
decrease of similar magnitude (-35%) was observed in Johannesburg, where a national lockdown was issued on 26
March 2020, with a gradual easing of restrictions starting 1 May. In Sub-Saharan Africa, the emission reductions in
April were significant for larger populous and industrialized areas, whereas no noticeable drop was found in less
developed regions (Masaki et al., 2020).
Finally, the Iraqi capital of Baghdad faced an initial lockdown from 22 March through 21 April. A second partial
lockdown was issued starting 20 May in response to a sharp increase in COVID-19 cases due to the temporary
relaxation of restrictions to allow the celebration of Ramadan in late April (Table C2). The $NO_2$ column responded
quickly (Figure 2n) as confirmed by the rapid decrease once curfew measures were issues in late-May.
Figure 3 and Figure 4 illustrate the tropospheric concentration of $NO_2$ over Europe, focusing on Milan, Madrid,
Paris and Berlin (Figure 3), extending the analysis to include summer months. In France, Spain and Italy we detect
strong reductions of $NO_2$, which can be largely attributed to the lockdown measures. In Berlin, the measured
differences are smaller, and a more detailed analysis of the meteorological variability is needed to quantify the impact
of the lockdown (see Figure 3). The extended time series shows a recovery of the $NO_2$ pollution levels to pre-COVID-
19 values. However, the recovery is not complete, suggesting that remaining restrictions, new stay-at-home life and
working practices, together with a downturn in industrial and service-based activities have contributed to a longer
lasting impact.


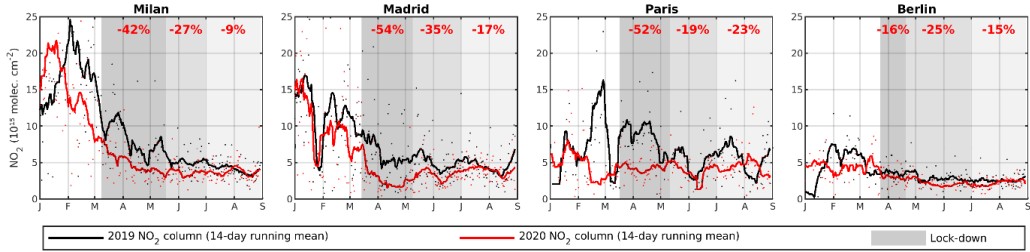


**Figure 3: Same as Figure 2, for the European cities Milan, Madrid, Paris, and Berlin, for an extended period of 1 January to 1 September. Additional shading indicates the lockdown period (dark grey), a transition period (grey), and the period with relaxed regulations (light grey).**


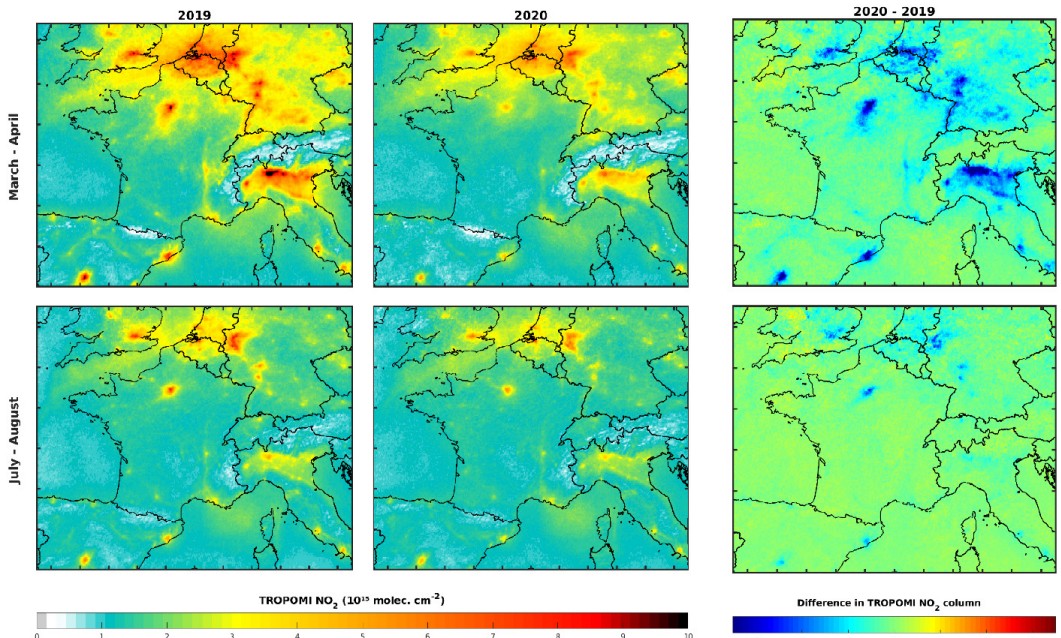


**Figure 4: TROPOMI NO₂ tropospheric columns over Europe in the lockdown months March-April (top) and the post-lockdown months July-August (bottom), comparing 2019 (left) with 2020 (middle). The difference is shown in the right panel.**


Relative concentration changes between 2019 and 2020, as mentioned previously, should not be fully attributed to COVID-19 lockdown measures and the subsequent reduction of emissions. Daily changes in the weather have a strong influence on the $NO_2$ concentrations, even when the data is averaged over a month. In order to estimate the impact of meteorological variability on TROPOMI-based $NO_2$ observations, simulations were performed with the LOTOS-EUROS chemistry-transport model over Europe at a resolution of 0.1° x 0.1°. Using the same emissions for 2019 and 2020, the simulations show that meteorological variability is responsible for changes in the monthly-mean, city-





averaged $NO_2$ columns with a 1-sigma standard deviation of about 13%. This variability is clearly illustrated in e.g.
the individual daily observations in Figure 2. The drastic changes in the range of 30-60% observed in the TROPOMI
data and shown in Figure 1 through Figure 4 clearly fall outside this range and cannot be attributed to weather alone.
A second complication is the presence of clouds. Months with persistent local cloud cover will therefore have a
reduced number of tropospheric column observations and will exhibit more natural variability. For quantitative
estimates of the COVID-19 measures, these factors should be carefully taken into account. This can be done through
(i) daily-based analysis of the $NO_2$ plumes from cities using wind speed fields from meteorological models and
subsequent emission derivation (Lorente et al., 2019; Goldberg et al., 2019); (ii) regression models to estimate the
impact of natural variability and emission trends in the observations (Diamond and Wood, 2020); (iii) chemistry-
transport modelling (Chang et al., 2020; Liu et al., 2020; Barré et al., 2021); and (iv) inverse modelling and data
assimilation approaches  (Ding et al., 2020; Miyazaki et al., 2020).
**4    Regional Observations for China**
China was the first country to impose measures to limit the spread of the SARS-CoV-2 virus. Although no national
lockdown was declared, strict local lockdown measures were implemented in many cities and provinces. In Wuhan,
the epicenter of the virus outbreak, the lockdown period lasted from 23 January 2020 until 8 April 2020, while in other
regions, it generally started in early February with measures being eased and lifted through March. In addition to the
lockdown measures, the yearly Chinese New Year holidays also affected the amount of anthropogenic emissions (Tan
et al., 2009), and so needs to be considered for proper interpretation of the observations. The timing of the holiday
period differs from year to year and took place from 24 January to 2 February in 2020, and in the periods 4-10 February
and 15-21 February for 2019 and 2018, respectively.
The impact of the COVID-19 crisis on air quality in China has already been investigated in several studies. Bauwens
et al. (2020) reported that tropospheric $NO_2$ column amounts observed by TROPOMI during the lockdown dropped
by 40-50% in the most impacted cities compared to the same period in 2019 (see Sect. 3). Accordingly, top-down
estimated NOx emissions exhibited sharp reductions of up to 50% during the strict lockdown period in late January
through early February (Ding et al., 2020; Liu et al., 2020; Zhang, R. et al., 2020).
In situ data indicate significant reductions of ground concentrations for $NO_2$, but also for PM, $SO_2$, and CO (Shi and
Brasseur, 2020; Wang et al., 2020; Zhang, Z. et al., 2020; Zhao, Y. et al., 2020). On the other hand, those studies
consistently reported increases of ozone concentrations. With the support of models, Zhao, Y. et al. (2020) have shown
that the observed decreases in $NO_2$ concentration were mostly caused by emissions reductions. They also show that
the contribution of meteorological changes to the observed concentration reductions of other species depends on the
exact location. Based on OMI observations, Zhang, Z. et al. (2020) observed reductions in East Asia of about 33%
and 41% for $NO_2$ and $SO_2$, respectively.
City-scale impacts of lockdown on $NO_2$ tropospheric column amounts for Wuhan and Beijing in Sect. 3. Here, we
investigate whether a lockdown signature can be detected from space at the regional scale for other key pollutants by
focusing on TROPOMI tropospheric column measurement of $SO_2$, CO, HCHO, and CHOCHO. We also compare the
identified changes with the marked changes in $NO_2$ concentration. Figure 5 compares monthly mean tropospheric





columns of those different species for February 2019 and 2020. The $NO_2$ and $SO_2$ tropospheric column amounts are
clearly lower in February 2020 compared to 2019. A small general reduction is also visible in the CO, HCHO and
glyoxal column amounts. As discussed before, many factors other than the lockdown measures may explain changes
in pollutant concentrations, such as the meteorology or emission reduction related to the timing of holidays. Another
difficulty to compare different years is the data sampling. In February 2019, large parts of Southern China were
covered by clouds, preventing space-based observation of the lowermost atmospheric layers. This is clearly illustrated
in the upper panel of Figure 5 showing CHOCHO concentrations, where data is missing over large regions since this
product uses the most stringent cloud filtering as compared to the other trace gases. Therefore, the following detailed
discussion only focuses on the northern part of China (black box in Figure 5 top left panel), even though the lockdown
measures were stricter in the region of Wuhan.


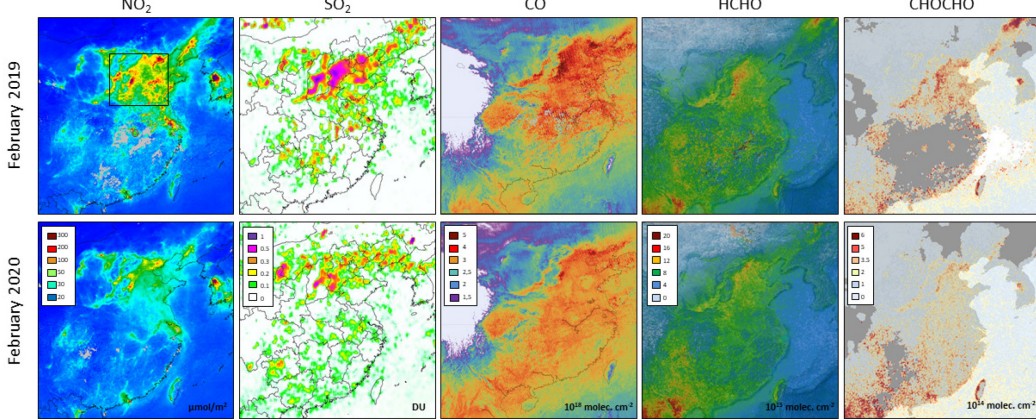


**Figure 5: Tropospheric and total columns for various trace gases over China as observed by TROPOMI over China in**
**February 2019 (upper panels) and 2020 (lower panels). The black box indicates the geographical region used in the time**
**series analysis (Figure 6). Note: the grey-shaded regions in $NO_2$ and CHOCHO panels (far left and far right, respectively)**
**indicate areas with little or no data available due to persistent local cloud cover.**


Figure 6 shows the seasonal cycles for tropospheric column amounts of TROPOMI $NO_2$, $SO_2$, CO, HCHO, and
CHOCHO for different years in northern China (region in black box highlighted in Figure 5) starting at the beginning
of the operational phase of the S5P/TROPOMI mission (30 April 2018). The different colored curves show two-week
medians of the daily mean tropospheric columns. In order to focus on the effect of COVID-19 lockdown measures for
HCHO and CHOCHO, the TROPOMI-based time series are compared with an OMI-based climatology for these
species using OMI data from 2010 to 2018, and shown by the black dashed curves. The associated uncertainties
represent the interannual variability as estimated from OMI. This type of climatological reference based on a longer
time series is not available for CO. Therefore, Figure 6 shows CO columns starting from 1 January 2018, which have
been added to extend the time series even though the data sampling was more limited in the early phase of the mission.





The light vertical boxes in January and February indicate the period of Chinese New Year holidays. Note that the 2020
holiday period was slightly extended as a first measure against the COVID-19 spread.


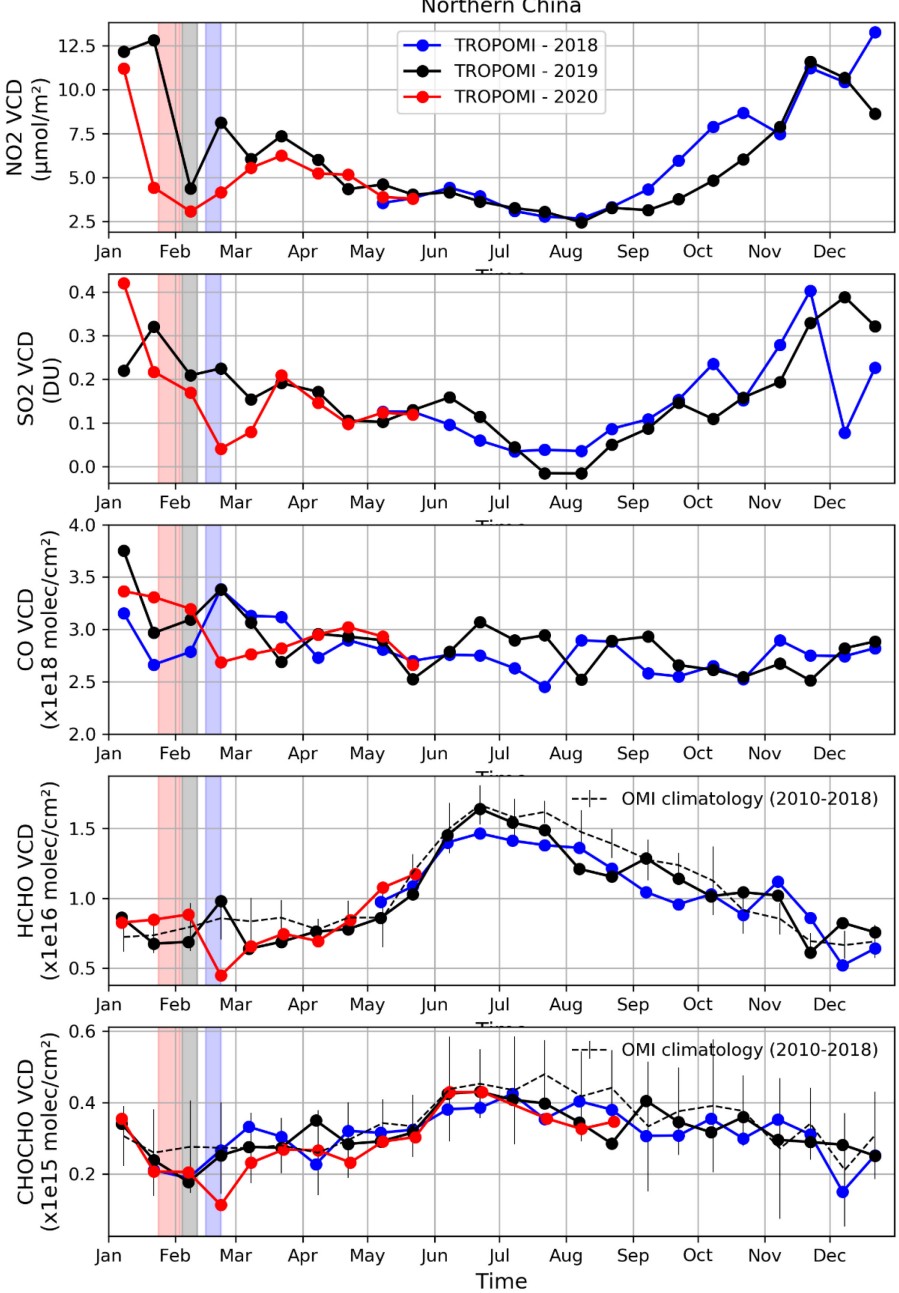




**Figure 6: Two-week median tropospheric column concentrations of NO₂, SO₂, CO, HCHO and CHOCHO (from top to**
**bottom) for northern China (34°N-40°N; 110°E-120°E). The different curves represent different years as indicated in the**
**legend. The colored boxes represent the yearly Chinese New Year holidays for those same years. The dashed black lines in**
**the HCHO and CHOCHO panels represent a climatological seasonality as obtained using the OMI data sets from 2010 to**
**2018 and the error bars represent the interannual variability (1-sigma standard deviation).**

Superimposed on the overall seasonal cycle of $NO_2$ (maximum during wintertime caused by a longer atmospheric
lifetime), a clear reduction of the $NO_2$ columns is systematically observed which corresponds to the New Year
festivities. While a quick return to higher values is usually observed after that period (Tan et al., 2009), the $NO_2$
columns remained lower for several weeks in 2020 likely as a consequence of the reduced traffic and industrial
activities. For example, $NO_2$ column amounts at the end of February were about 45% lower than those of 2019. In
March 2020, $NO_2$ columns return progressively to a similar level as compared to other years.
$SO_2$ emissions in China mostly originate from fossil fuel burning of coal and oil (Wang et al., 2018). Although
Chinese $SO_2$ emissions have dropped significantly in the last decade (van der A et al., 2017; Zheng et al., 2018a),
enhanced $SO_2$ columns are still observed in some regions of northern China (Figure 5). As illustrated in Figure 6, $SO_2$
column amounts are larger during wintertime mostly due to its longer atmospheric lifetime (Lee et al., 2011). No clear
reduction could be related to the yearly holidays. However, in 2020 a sharp drop is observed starting in late January
through mid-March with a reduction of up to 77% as compared to 2019. By late-March/early-April values returned to
levels similar to previous years, which is consistent with the $NO_2$ lockdown signature.
In northern China the residential sector, consisting of mostly of emissions from heating and cooking, accounts for
nearly half of the anthropogenic CO emissions, while the rest is distributed between traffic, power generation, and
industry (Zheng et al., 2018b). Since the impact of lockdown measures is more limited for the residential sector as
compared to the transport or industrial sectors, the response of CO to the lockdown measures is expected to be less
distinct. Also, due to the longer atmospheric lifetime of CO (weeks to a month), the observed column amounts result
from the accumulation of the trace gas over source regions and from long-range transport from regional and global
sources. As such, meteorology significantly influences CO concentrations. The observed day-to-day variability is
indeed large, leading to more scatter in the two-week median time series shown in Figure 6. The CO columns observed
in late February/early March are lower than those observed in the last two years, which might be partly caused by the
lockdown measures. However, the high temporal and spatial natural variability of the CO column amount is of the
same magnitude as the possible COVID-19 lockdown signal, and the large, year-to-year interannual differences
prevent firm conclusions from being drawn. Dedicated model simulations or a longer time series of the TROPOMI
CO data may help to disentangle these effects in the future.
There are difficulties associated with the investigation of a possible lockdown signature in the satellite HCHO and
CHOCHO data sets. Large uncertainties are associated with both of these column retrievals owing to their low optical
depth. Moreover, HCHO and CHOCHO columns are dominated by biogenic emissions, which explains the observed
seasonal pattern of HCHO and CHOCHO column values with a maximum during summertime as illustrated in Figure
6. Variability in meteorology (temperature changes, winds, precipitation) may lead to changes in column amounts on
the same order of magnitude as the expected lockdown-related reduction in anthropogenic emission changes. The
interannual variability as inferred from the OMI data sets is estimated to be in the range of $1 \times 10^{14}$ molec cm⁻² (~30%)





and $1.2 \times 10^{15}$ molec cm$^{-2}$ (~12%) for CHOCHO and HCHO, respectively. Despite those issues, a clear minimum is
visible for both HCHO and CHOCHO in late February 2020, with columns significantly lower than 2019 and lower
than the OMI climatology (about -40% and -50% for HCHO and CHOCHO, respectively). The differences are also
larger than what can be explained by the typical interannual variability. This is in agreement with Sun et al. (2021),
who finds a significant HCHO decrease in the Northern China Plain. For glyoxal, a reduction of the column amounts
starts already in late January but similar reductions are observed in other years and might be related to a holiday effect
similar to that observed for NO$_2$.
It is interesting to note that local minima are observed simultaneously in late February 2020 for all species except
NO$_2$, despite the data products being generated using independent retrieval algorithms. This gives confidence into the
detected reductions and their anthropogenic origin. The small delay between the initial decrease in NO$_2$ concentration
and the observed decreases in the other trace gas signals is related to a combination of longer atmospheric lifetimes
and production being dominated by secondary processes as compared to NO$_2$ and is also likely tied to the early timing
of the Chinese New Year in 2020.
**5  Regional Observations for India**
India implemented strict national lockdown measures limiting activities across the country starting 24 March 2020 for
a period of 21 days in order to tackle the spread of the SARS-CoV-2 virus amongst its 1.3 billion inhabitants. The
initial stringent phase 1 restrictions were followed by careful region-based relaxations in three subsequent phases
carried out through the end of May as shown in Table 2.

**Table 2: Lockdown phases in India.**

|         | Dates          | Measures                                                                                                               | Reference                   |
|---------|----------------|------------------------------------------------------------------------------------------------------------------------|-----------------------------|
| **Phase 1** | 24 Mar to 14 Apr | Nearly all services and factories suspended.                                                                           | Singh et al. (2020)         |
| **Phase 2** | 15 Apr to 3 May  | Extension of lockdown with relaxations, reopening of agricultural businesses and small shops at half capacity.         | BBC News (2020)             |
| **Phase 3** | 4 May to 17 May  | Country split in 3 zones: (i) lockdown zone, (ii) zone with movement with private and hired vehicles, and (iii) normal movement zone. | India today (2020)          |
| **Phase 4** | 17 May to 31 May | Additional relaxations, more authority given to local bodies.                                                          | The Economic Times, 2020    |



Figure 7 gives an overview of TROPOMI observations of $NO_2$, $SO_2$, CO, HCHO, and CHOCHO, over India for
April 2020, thus covering most of phase 1 and 2 of the Indian lockdown, as compared to the same month in 2019. For
$NO_2$ and $SO_2$ the concentrations are clearly lower across the country in 2020 as compared to 2019. Although less
prominent, concentrations of CO, HCHO, and CHOCHO appear to be lower in April 2020 over the domain of the
Indo-Gangetic Plain (IGP), which is one of the most densely populated areas of the world with roughly 900 million
people.

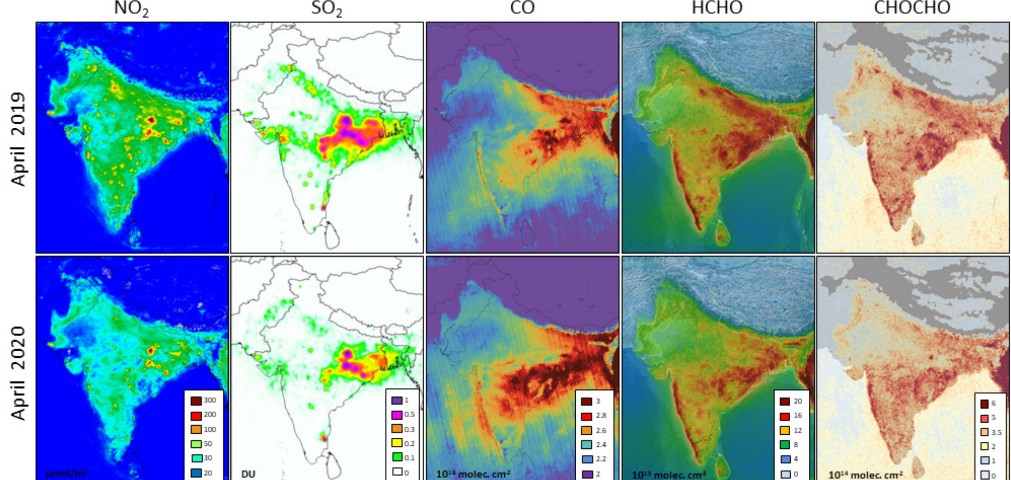

**Figure 7: Concentrations maps for April 2019 (top row) and April 2020 (bottom row) for the various trace gas species**
**measured by TROPOMI from left to right, $NO_2$, $SO_2$, CO, HCHO and CHOCHO.**

The two main sources of $NO_2$ are road transport and power generation, each accounting for about 30% of total
anthropogenic emissions in India (Granier et al., 2019). During phase 1 of the lockdown the Tom-Tom traffic index
dropped by 80% (Aloi et al., 2020; Prabhjote, 2020) and energy consumption dropped by 25% compared to 2019
(Dattakiran, 2020; POSOCO, 2021) (Fig. D1). As such, we expect a strong reduction in $NO_2$ particularly in urban
areas due to large decreases in transport sector activities and we expect a weaker reduction near power plants due to
smaller decreases in energy demand.
Indeed, as indicated by the maps of $NO_2$ column concentrations in Figure 7, a notable reduction in $NO_2$ can be seen
in April 2020 as compared to April 2019. A clear reduction is observed over major cities as well as over the eastern
part of India where most large power plants are located. Figure 8a shows the average $NO_2$ total column concentrations
as measured by TROPOMI for 2018, 2019 and 2020, for the 40 largest cities in India selected on the basis of the
number of inhabitants (www.geonames.org) where $NO_2$ is averaged over a 15 x 15 $km^2$ area around each city center.
When both city centers and power plants are located within a 45 x 45 $km^2$ box, this box is excluded from the averages
to avoid potential outflow of one source to the other. A sharp reduction of 42% can be seen in the amount of $NO_2$ over





cities during the first phase of the lockdown period starting at the end of March, as compared to the same period in
2019. This initial drop in $NO_2$ is then followed by a slow but gradual increase in line with the successive relaxation
phases (Table 2). Power generation is a major source for $NO_2$ in India, in particular from coal-fired power plants.
When examining the average amount of $NO_2$ over the 100 largest coal-fired power plants (www.wri.org), we observe
a significant drop in $NO_2$ during phase 1 of the lockdown period. This drop, observed over coal-fired power plants of
23% as compared to 2019 (Figure 8b), is less pronounced than the observed drop in $NO_2$ over cities (Figure 8a). The
TROPOMI-observed reduction in $NO_2$ over coal-power plants is in line with the initial 25% decrease in maximum
electricity demand reported by National Load Dispatch Centre (NLDC) during phase 1 and tapering to an 8% decrease
during phase 4 of the lockdown as compared to 2019 (Fig. D1, Dattakiran, 2020).

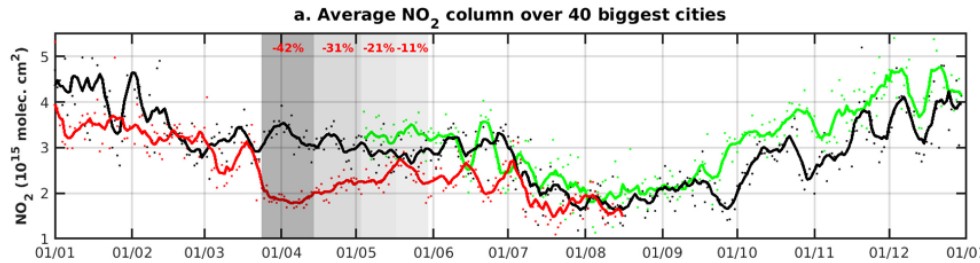

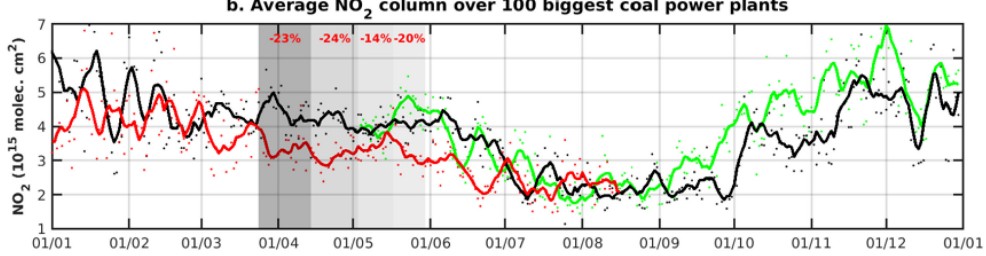

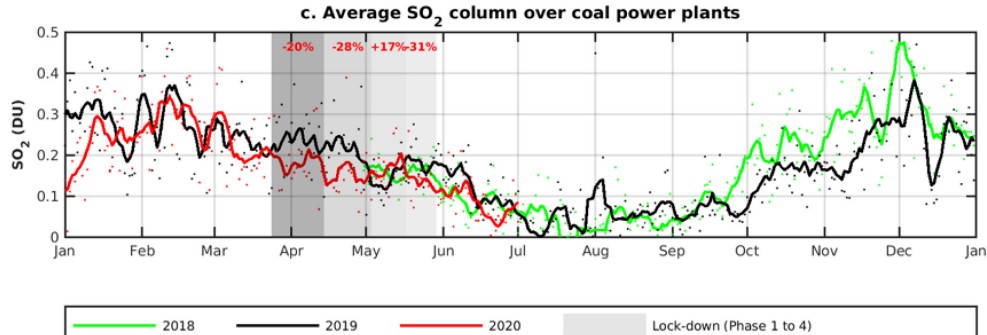


**Figure 8: Average tropospheric $NO_2$ concentrations for May 2018 (green), 2019 (black) until June 2020 (red) over the 40**
**largest Indian cities (top); over the 100 largest power plants in India (middle); and average $SO_2$ concentrations over the**
**59 largest $SO_2$-emitting power plants in India (bottom). The four different phases of the lockdown period are denoted by**



**the different grey shading. For each phase, the reductions in NO$_2$ (or SO$_2$) concentrations are given relative to the same**
**period in 2019. The dots are the daily means, and the solid lines represent the 7-day running means.**

According to the CAMS-GLOB-ANT emission inventory for 2019 the major sources for SO$_2$ in India are power
generation (65%) and industry (25%) (Granier et al., 2019). Since India largely relies on coal for producing energy, it
is the world's top emitter of anthropogenic SO$_2$ (Li et al., 2017). So, most of the SO$_2$ signal we see in TROPOMI data
for this region (Figure 7) is from coal-fired power plants, where contributions from oil and gas plants in India comprise
a much smaller part of the signal (Fioletov et al., 2016). From Figure 7, a reduction in SO$_2$ is visible over most areas,
and is especially noticeable for the easternmost part of India, which is India's largest SO$_2$-emitting region with more
than 20 coal-fired power plants.
We have investigated the SO$_2$ VCD amounts over the largest power plants, and adapted the selection method used
for NO$_2$ by considering a larger area of 50 x 50 km$^2$ around each power plant. This is justified by (1) the longer lifetime
of SO$_2$ compared to NO$_2$, (2) the lower contamination by other sources, and (3) the need to reduce the noise on the
SO$_2$ data to more clearly isolate the signal from the power plant. The results of the averaged SO$_2$ VCD time series are
presented in Figure 8c. It should be noted that, compared to NO$_2$, an additional selection of the power plants was
applied. Based on the SO$_2$ VCD map for April 2019 (Figure 7), only the power plants with mean SO$_2$ columns larger
than 0.15 DU were considered (59 power plants in total). Although the signal is relatively weak for SO$_2$, we find very
similar reductions in SO$_2$ as compared to NO$_2$. Especially during the first two phases of the lockdown, a reduction of
about 20% is found which is in line with the NO$_2$ observations and the reported reduction in energy demand. In May,
for the different years, the consistency between NO$_2$ and SO$_2$ VCDs is less straightforward and the reason for this is
not fully understood. It should however be noted that the NO$_2$ and SO$_2$ data products do not use the same cloud
products for filtering and this might be a reason for discrepancy. Moreover, the possibility of a systematic
contamination of the NO$_2$ signal over power plants by other sources cannot be ruled out completely. A noticeable
feature of Figure 8b and Figure 8c is the overall excellent correspondence between NO$_2$ and SO$_2$ VCD evolution (on
short-term/seasonal basis, and outside the lockdown periods) as well as from year to year. This further strengthens the
observed COVID-19 related drop in both trace gases, although it is clear that meteorology and chemistry likely play
a large role in the observed VCD variability. Also, ground-based studies in New Delhi find a more important reduction
in NO$_2$ compared to SO$_2$ (Mahato et al., 2020; Kumari and Toshniwal, 2020).
For HCHO, CHOCHO, and CO, various regions over India have been investigated to detect a possible signal
resulting from COVID-19 lockdown measures. We could only identify such a signal in the densely populated areas of
the Indo-Gangetic Plain and New Delhi. These areas, due to the high intensity of traffic and industrial activities, are
most likely to exhibit large impacts on atmospheric pollution levels due to COVID-19 lockdown measures.
Figure 9 shows two-week averaged column values for HCHO, CHOCHO, and CO over the IGP and New Delhi,
based on TROPOMI data from January 2018 to June 2020. To support the interpretation of the observed seasonal and
interannual variations, Fig. D2 presents the corresponding temperature, precipitation amount, and fire count. The
temperature starts increasing in January and reaches a maximum in June. The period from July to September
corresponds to the monsoon season with heavy rains and lower temperatures, and therefore lower pollution levels.
Fire activity peaks around May with a second peak is observed in November for the IGP. The time series of the HCHO,





CHOCHO, and CO columns correlate with these seasonal events, although with a different amplitude. For example,
HCHO shows the strongest correlation with temperature (see Sect. 2.5), while CHOCHO mainly follows fire
emissions. The smaller amplitude in CO variations is caused by its longer lifetime.


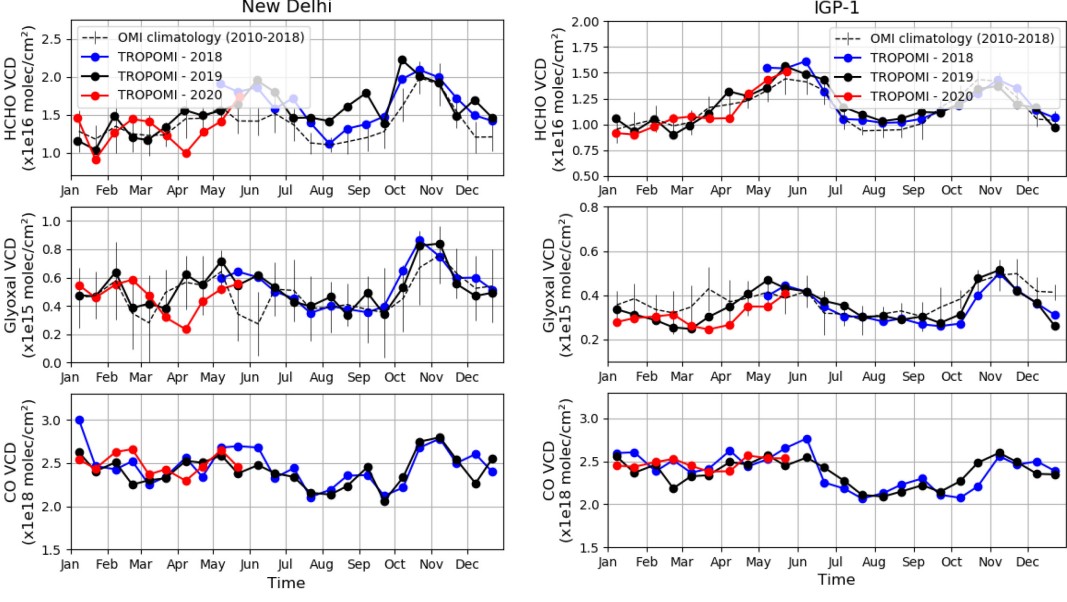


**Figure 9: Time evolution of HCHO, CHOCHO, and CO over the densely populated Indo-Gangetic plain (defined by the region within this 4 coordinates: 29.5°N 72°E, 21.5°N 86°E, 24.5°N 88.5°E , 32.5°N 74.5°E), and over the megacity New Delhi (radius of 25 km, or 50 km for CHOCHO) as observed with TROPOMI. The year 2020 is represented in red (2018 in blue, 2019 in black). With the HCHO and CHOCHO time series, the OMI climatology is shown for comparison (dashed black line, 2010-2018), the error bars represent the interannual variability of the two-week averaged columns. The HCHO columns have been corrected in order to assume the same temperature every year (see Sect. 2).**


A large part of the observed HCHO and CHOCHO columns for India are due to natural emissions which can vary
significantly due to changes in meteorology, in particular temperature and precipitation. Hence a possible reduction
of the anthropogenic VOC emissions due to the lockdown measures is expected to have a small contribution to the
variability of the measured columns. During the most stringent phase 1 lockdown, a reduction in HCHO column
concentrations is observed for the IGP and is even more pronounced over New Delhi (Figure 9 top panels; respectively
-2 and -4x$10^{15}$ molec cm$^{-2}$ [-20% and -40%] compared to the OMI climatology for 2010-2018). In both cases, the
anomaly is larger than the interannual variations observed during this period (about 1.5x$10^{15}$ molec cm$^{-2}$), where
changes in temperature or precipitation do not seem to explain the observed column decrease during phase 1. The





observed column decline is even more pronounced over New Delhi than over the IGP, suggesting that the origin of
the reduction is mostly anthropogenic.

729       The case for lockdown-driven reductions is further supported by the CHOCHO observations, which exhibit the

clearest COVID-19 signal during phase 1 of the lockdown (Figure 9). The reduction of CHOCHO during the lockdown
period over the IGP is slightly larger than the interannual variability of $1 \times 10^{14}$ molec cm$^{-2}$ (or -25%) as determined
from the OMI CHOCHO climatology. Similar to HCHO, the reduction in CHOCHO over New Delhi is twice as large
(-50%) and well beyond the 1-sigma OMI climatology range. Phase 2 is also characterized by lower CHOCHO column
amounts in 2020 as compared to 2019, but temperatures are also lower, unlike phase 1. Accounting for temperature-
driven variability (Sect. 2.5) brings the HCHO columns close to the mean HCHO seasonal levels. The somewhat more
pronounced effect of the lockdown on CHOCHO compared to HCHO in New Delhi is most likely due to the strong
contribution of anthropogenic VOC precursors to CHOCHO amounts (Chan Miller et al., 2016). Interestingly, fire
counts show that there were fewer fires in May 2020 compared to previous years (Fig. D2), most likely as a
consequence of the lockdown measures, which may also contribute to the lower glyoxal columns.

740       As it was the case for China, it is more difficult to identify a signal in CO column data driven by the COVID-19

lockdowns over India. An important reason for this is the much longer atmospheric residence time of CO that varies
depending on the OH concentration (Holloway et al., 2000). Moreover, according to bottom-up inventories, the major
anthropogenic CO source in India are due to the residential sector (42%), road transportation (21%), agricultural waste
burning (18%) and the industrial sector (16%) (Granier et al., 2019). Hence, during a lockdown we expect that the
main source of CO, residential, to be less affected. Figure 7 shows that the CO amounts in southern India are higher
in 2020 compared to 2019. The reason could be the accumulation of CO originating from elsewhere prior to the
lockdown period. The long atmospheric residence time of CO complicates the identification of COVID-19 lockdown
signals. Also for CO we derived the full TROPOMI time series for the IGP and New Delhi as shown in Figure 9
(lower panel). The time series for New Delhi in mid-April shows somewhat lower CO values in 2020 compared to
2019, but the large natural variability of CO prevents clear identification of a COVID-19 lockdown driven effect. In
future, analysis of a longer TROPOMI CO time series or model experiments may help to quantify the COVID-19
effects.

## 754    6    Conclusions

In this paper, we have analyzed the impact of COVID-19 lockdown measures on air quality around the globe, based
on observations of several trace gases from the Sentinel-5P/TROPOMI instrument. TROPOMI provides daily, global
observations of multiple trace gases, where the measured vertical column amounts are driven by emissions as well as
atmospheric and chemical processes of transport, transformation, and deposition. We compared the 2020 TROPOMI
data with similar periods from previous years and carried out additional analysis to disentangle changes in emissions
due to COVID-19 lockdown measures from meteorological variability, seasonal variability, and from other non-
lockdown emission drivers. We analyzed time series of NO$_2$ measurements from city to regional scales for several
locations around the globe, showing the potential of TROPOMI to globally monitor local to regional impacts of



COVID-19 lockdown measures on air quality and anthropogenic emissions. Furthermore, for the first time, we used
a combination of five trace gases observed by TROPOMI, specifically $NO_2$, $SO_2$, CO, HCHO and CHOCHO, to assess
the impact of COVID-19 related lockdown measures on trace gas concentrations.
From the global to city scale, we have illustrated consistent, sharp decreases in $NO_2$ concentrations driven by the
COVID-19-related lockdown measures. These findings are based on detailed analysis of the distribution of $NO_2$ using
daily measurements from TROPOMI. For the city of Wuhan in China, the first city to issue a lockdown, $NO_2$
concentrations measured by TROPOMI were about 60% lower than the same period in February-March 2019. After
China, lockdowns were issued across all continents and for the majority of countries from March through May 2020.
For megacities all over the world, reductions in column amounts of tropospheric $NO_2$ range between 14% and 63%.
The strength of the reduction depends on the type and efficiency of local measures carried out and on the relative
contribution of traffic, industry, and power generation to $NO_2$ emissions for a given area. Owing to the unprecedented
resolution of TROPOMI of about 5 km, reductions of different source contributions to $NO_2$ such as city traffic,
highways (Liu et al., 2020), power plants (Miyazaki et al., 2020), industry, and shipping (Ding et al., 2020) can be
estimated separately.
As demonstrated by time series analysis of the $NO_2$ observations, there is substantial variability even in two-week
averages, which is attributable to meteorological variability. On average, we estimate the standard deviation of this
variability to be about 13% (1-sigma standard deviation) for major cities in Europe, but locally the effect can
sometimes be larger. The large and systematic reductions (30-60%) observed, however, cannot be explained by
meteorological variability alone and are therefore attributed to the effect of the lockdown measures.
For $SO_2$, we observe significant column reductions in China and India over coal-fired power plants, which are the
primary sources of anthropogenic $SO_2$ in these areas. Over northeastern China in late February 2020, large reductions
of $SO_2$ vertical column amounts were observed, as a result of lockdown measures, with a decrease up to 77% as
compared to the same time period in 2019, which cannot be explained by interannual variability alone. An analysis of
$SO_2$ vertical column amounts over the largest $SO_2$-emitting power plants in India, reveals a reduction in $SO_2$ of about
25% during the first two phases of the lockdown, as compared to 2019. For India, the reductions in $SO_2$ were highly
correlated with $NO_2$ reductions for the same power plants and with the national energy demand for that period.
The natural variability of HCHO and CHOCHO does not allow detection of a significant decrease due to the COVID-
19 measures in most regions of the world based on TROPOMI observations alone. Exceptions are northern China and
New Delhi, where observed reductions could be attributed to the lockdown measures. For northeastern China, a 50%
reduction in the CHOCHO concentration is observed during the second half February, which is larger than the typical
observed interannual variability of 30%. For HCHO, after correcting for the effect of seasonal and temperature
variations, we observe a coincident 40%. We analyzed column amounts of CO, CHOCHO, and HCHO over the Indo-
Gangetic Plain, which is the most densely populated region of India. For CHOCHO and HCHO, we observed small
reductions in column amount due the COVID-19 measures, where these observed effects are slightly larger than the
interannual variability as determined using an OMI climatology (2010-2018). The observed reduction of 25% of
CHOCHO in this region is of the same order as the typical interannual variability. A stronger reduction of 60% is
observed for the city of New Delhi, which is similar to the reduction observed over northern China but occurs later





due to the difference in lockdown timing. For HCHO, we also observe a significant 40% decrease over New Delhi in
April, while over the whole Indo-Gangetic Plain, a decrease of 20% is observed.
For CO, reductions related to COVID-19 measures were much more difficult to identify, although over northern
China we see that the reductions in CO correlate with those for HCHO and CHOCHO. We could not find a similar
effect for CO over New Delhi. The fact that it is so hard to draw conclusions for CO based on the TROPOMI data
alone is due to the high variability in CO driven by meteorological conditions, in combination with the difficulty of
distinguishing localized emission changes from the high and variable background values, caused by the long
atmospheric lifetime of CO.
TROPOMI data have already been used in many publications (Gkatzelis et al., 2021; Bauwens et al., 2020; Liu et
al., 2020; Huang et al., 2020) aiming to analyze the impact of COVID-19 lockdown measures on air pollution levels.
Predominantly, these studies have been based on the use of TROPOMI $NO_2$ observations alone. We anticipate that
the combined use of multiple trace gases from TROPOMI together with the high spatial resolution of the
measurements, has large potential for a significantly improved sector-specific analysis of the impact of the COVID-
19 lockdown measures than previously possible. Such a multi-species analysis offers promise for in-depth
understanding of changes in air quality, the chemical interplay of pollutants in the atmosphere and their relation to
emissions. While keeping in mind the importance of accounting for interannual, seasonal, and meteorologically driven
variability (e.g. Miyazaki et al., 2020), it is clear that a detailed analysis cannot be based on TROPOMI observations
alone. For more quantitative estimates of the impact of COVID-19 lockdown measures on trace gas concentrations
and emissions, we need (inverse) models driven by high-quality meteorological analyses, or at least wind information
or statistical relationships to account for weather-driven variability (Goldberg et al., 2020; Miyazaki et al., 2020; Ding
et al., 2020).
In summary, our analyses using the most recent operational and scientific retrieval techniques have shown that by
taking emission sources, atmospheric lifetime as well the seasonal and meteorological variability into account for a
variety of trace gases measured by TROPOMI, rapid changes in anthropogenic emissions can be observed as induced
by the implementation of regional COVID-19 lockdown measures. It is our hope that this case study will serve as
reference for future analyses aimed at characterizing emission changes of not just $NO_2$, but by utilizing the
concomitant observation of the variety of trace gases measured by TROPOMI.

**Appendix A**
**Table A1: Summary of documentation available for TROPOMI operational data products from the Sentinel 5-P Library**
**(https://sentinels.copernicus.eu/web/sentinel/technical-guides/sentinel-5p/products-algorithms).**

| Title | Document content description and product-specific reference | Document and Data links |
| --- | --- | --- |
| **Product Readme File (PRF)** | Description of changes between different product versions and overall quality information | https://sentinels.copernicus.eu/web/sentinel/technical -guides/sentinel-5p/products-algorithms |



| | | |
|---|---|---|
| **NO₂** | Eskes and Eichmann, 2020 | |
| **CO** | Landgraf et al., 2020 | |
| **HCHO** | De Smedt et al., 2020a | |
| **Product User Manual (PUM)** | Technical description of file formatting for each TROPOMI Level 2 operational data product | https://sentinels.copernicus.eu/web/sentinel/technical-guides/sentinel-5p/products-algorithms |
| **NO₂** | Eskes et al., 2020 | |
| **CO** | Apituley et al., 2018 | |
| **HCHO** | Romahn et al., 2020 | |
| **Algorithm Theoretical Basis Document (ATBD)** | Detailed description of methods used for each TROPOMI L2 operational retrieval algorithm | https://sentinels.copernicus.eu/web/sentinel/technical-guides/sentinel-5p/products-algorithms |
| **NO₂** | van Geffen et al., 2019 | |
| **CO** | Landgraf et al., 2018 | |
| **HCHO** | De Smedt et al., 2020b | |
| **Quarterly Validation Report (ROCVR)** | Detailed description of the latest validation available for each TROPOMI L2 operational dataset, product-specific | https://mpc-vdaf.tropomi.eu/ |
| **Operational Data Product Specifications** | Product-specific overview pages with TROPOMI L2 dataset specifications, including how to access and how to cite each data product. | https://sentinels.copernicus.eu/web/sentinel/data-products |
| **Operational Data Product Citation and Digital Object Identifier (DOI)** | NO₂ Copernicus Sentinel 5-P, 2018a | doi:10.5270/S5P-s4ljg54 |
| | CO Copernicus Sentinel 5-P, 2018b | doi:10.5270/S5P-1hkp7rp |
| | HCHO Copernicus Sentinel 5-P, 2018c | doi:10.5270/S5P-tjlxfd2 |


**Appendix B**
Appendix B contains additional information supporting the timing of COVID-19 driven emissions changes for global
cities evaluated in this study and shown in Fig. 2.

**Table B2. Details about the lockdown dates for the cities illustrated in Figure 2.**





| City | Date (2020) | Comment | Reference |
|---|---|---|---|
| **Wuhan** | 23 January | Lockdown Wuhan and Hubei province | Bloomberg (2020) |
| | 8 April | Lockdown lifted | Bloomberg (2020) |
| **Mumbai and New Delhi** | 24 March | Closure of schools, public transport and most businesses | BBC (2020a) |
| | 31 May | Nationwide lockdown is extended until end of May | Aljazeera (2020a) |
| **Manila** | 16 March | Philippines announced strict home quarantine | Calonzo and Jiao (2020) |
| | 1 June | Most businesses allowed to re-open, but bars, restaurants and schools remain closed | Jennings (2020) |
| **Madrid** | 14 March | Nationwide lockdown | Minder and Peltier (2020) |
| | 9 May | Easing, stores and restaurants allowed to open | Goodman et al. (2020) |
| **Milan** | 8 March | Locking down of Northern Italy including Milan | Horowitz (2020a) |
| | 4 May | Loosening of strictest lockdown measures | Horowitz (2020b) |
| **Paris** | 17 March | France imposes nationwide the restriction | Onishi and Méheut (2020) |
| | 11 May | Gradually relaxed lockdown measures, most shops open | Makooi (2020) |
| **Los Angeles** | 19 March | California enters lockdown | BBC (2020b) |
| | 1 June | Reopening of some shops and restaurants | Patel (2020) |
| **New York** | 22 March | New York state enters lockdown | BBC (2020b) |
| | 13 June | Stay-at-home orders put in place until further notice | CBS News (2020) |
| **Sydney** | 24 March | Strict lockdown measures adopted in Australia | Wahlquist (2020) |
| | 15 May | New South Wales eases lockdown restrictions | Sonali (2020) |
| **Auckland** | 23 March | In New Zealand stay-at-home orders are issued | Menon (2020) |
| | 14 May | All businesses can open in New Zealand | Conforti (2020) |
| **Mexico City** | 23 March | Most economic sectors stopped in Mexico | Pasley (2020) |
| | 1 June | Gradual reopening of Mexico city | Associated Press (2020) |
| **Lima** | 16 March | Stringent quarantine enforced by police and army | Collyns (2020) |





| | | | |
|---|---|---|---|
| | 30 June | Peru extended nationwide lockdown through end of June | Aljazeera (2020b) |
| **Sao Paulo** | 24 March | Start of lockdown, but measures were largely ignored | Uchoa (2020) |
| | 31 May | Quarantine extended through May | CGTN (2020) |
| **Buenos Aires** | 20 March | Argentina under mandatory lockdown | Do Rosario and Gillespie (2020) |
| | 28 June | Lockdown extended | Misculin and Garrison (2020) |
| **Baghdad** | 22 March | Iraq imposed a total nationwide lockdown | The Star (2020) |
| | 21 April | Relaxed restrictions: shops reopen for limited hours | Saleh (2020) |
| | 20 May | In Baghdad strict lockdown re-imposed for 6 districts | Saleh (2020) |
| **Lagos** | 30 March | Stay-at-home order, markets open for limited hours | Orjinmo (2020) |
| | 4 May | Easing of restrictions, but schools, bars, and cinemas remain closed | Mbah (2020) |
| **Johannesburg** | 26 March | Stay-at-home orders issued in South Africa | Winter (2020) |
| | 1 June | Most economic sectors permitted to operate | Aljazeera (2020c) |



**Appendix C**

Appendix C contains figures which support the technical understanding of individual retrieval algorithms.




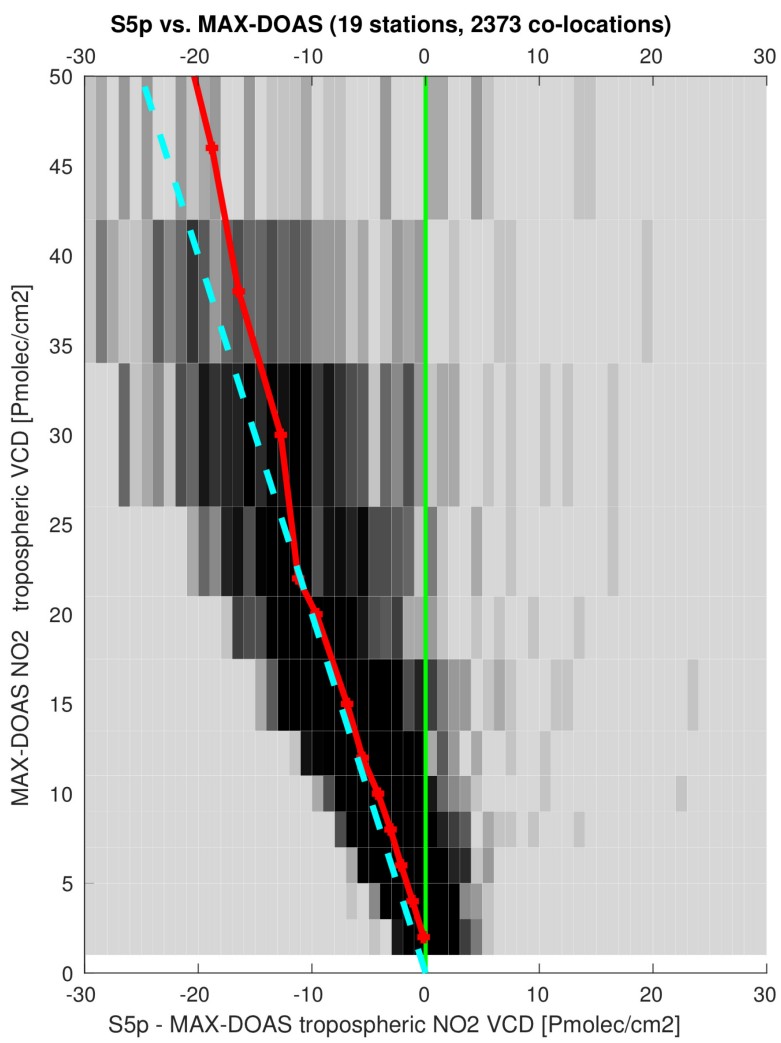

**Figure C1: Bias in S5p-TROPOMI tropospheric NO₂ as estimated from comparisons to co-located ground-based MAX-DOAS measurements, presented as a function of the ground-based VCD measurement. The grey-scale background represents a 2-D histogram, where the median difference per MAX-DOAS VCD bin is shown as the red curve, and the blue dashed line shows a multiplicative bias (b) model with b ~ 0.5 x VCD. More details on the ground-based data and co-location scheme can be found in Verhoelst et al., 2021.**





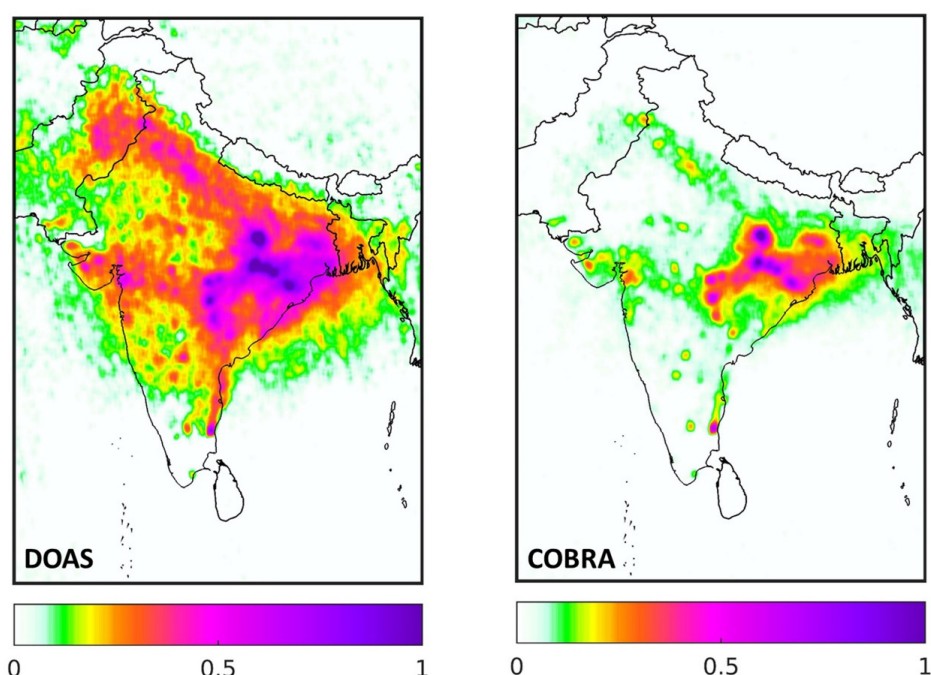

850

**Figure C2: Monthly averaged TROPOMI SO₂ columns over India for April 2019, from (left) DOAS operational product and (right) COBRA scientific product. The noise and offsets reduction is clear from the maps. The emissions from individual point sources (power plants) can be better discerned in the COBRA SO₂ map.**

854

855



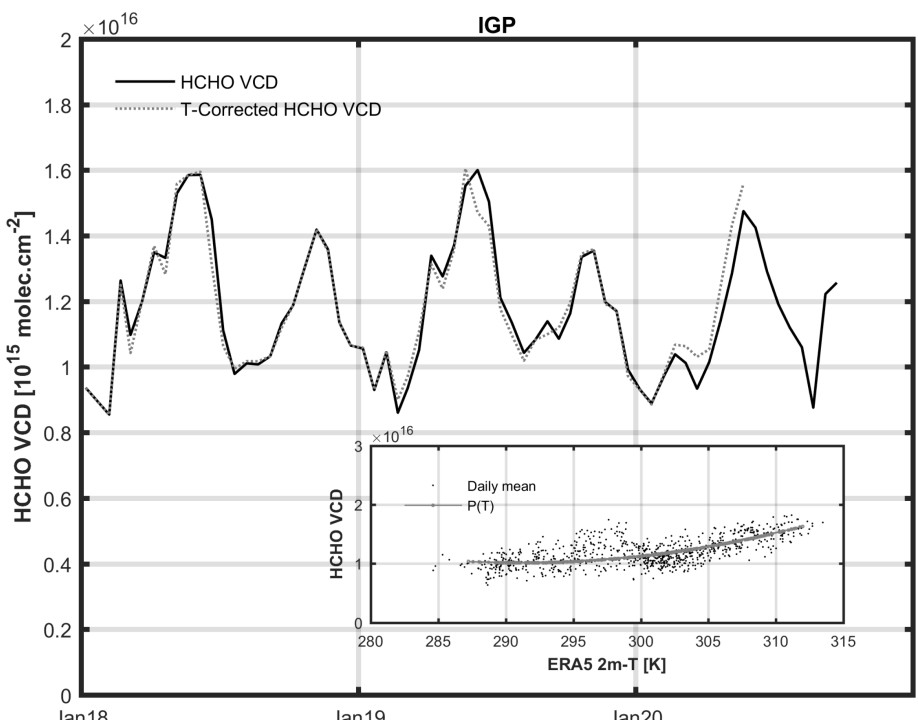

**856**

**857** Figure C3: Example of temperature correction of the TROPOMI HCHO tropospheric columns in the Indogangetic Plain
**858** region. The dashed line presents the HCHO columns after correction using climatological temperatures. The correlation
**859** between the local daily temperatures from ERA5-Land 2m and the HCHO columns is shown inset for the entire period.

**860** **Appendix D**

**861** Appendix D contains additional figures that support the interpretation timing of observed changes in COVID-19

**862** driven emissions related to power generation (Fig. D1) and meteorological conditions (Fig. D2).



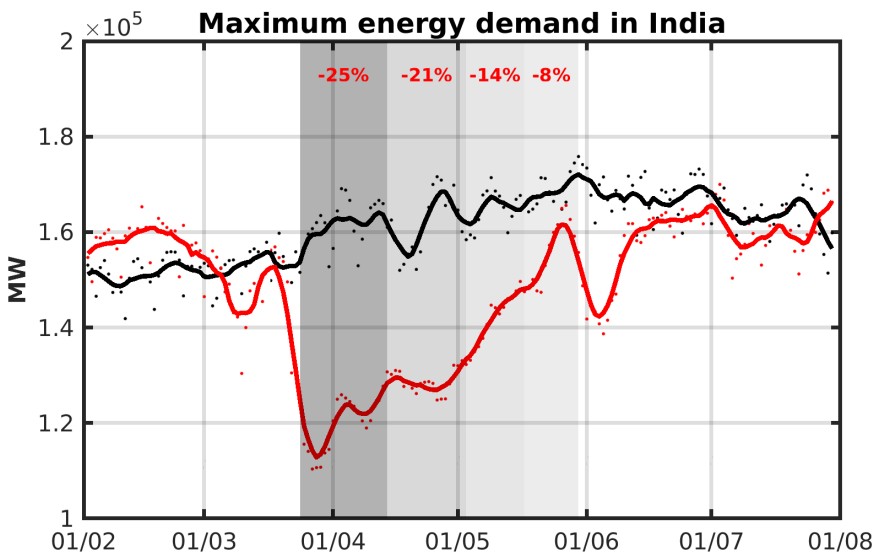


**Figure D1: Maximum energy demand over India during the period of the lockdown (red) compared to the same period in 2019 (black), For each of the phases of the lockdown the reductions in maximum energy demand is given relative to the same period in 2019.  Data from: www.posoco.in/covid-19.**




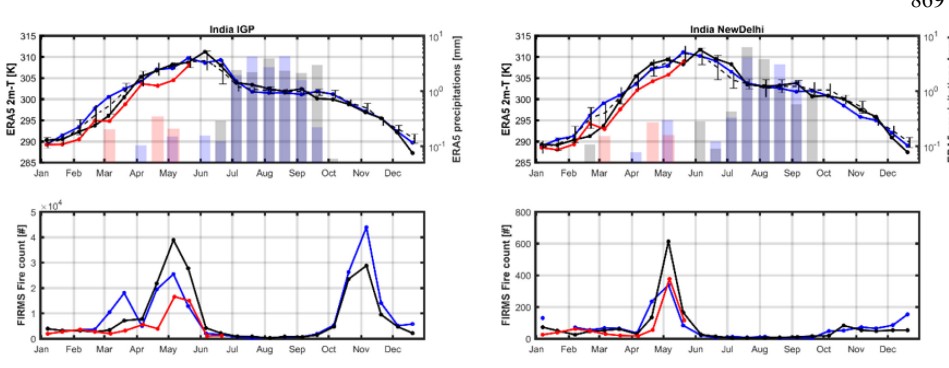

**Figure D2:  Upper panels: ombrothermic diagrams for the same regions as shown in Figure 9 showing the two-week
average temperature at 2m (upper left) and precipitation amounts (upper right, source ERA5, Muñoz Sabater, 2019b).
Lower panels: fire counts (source FIRMS, https://earthdata.nasa.gov/firms). The year 2020 is represented in red, 2019 in
black, and 2018 in blue.**



**Data Availability**

Operational versions of all Copernicus Sentinel 5-P Data TROPOMI data are freely available from the European Union/ESA/Copernicus Sentinel-5P Pre-Operations Data Hub (https://s5phub.copernicus.eu; S5P Pre-Ops Data Hub, 2021). The TROPOMI COBRA $SO_2$ dataset is available on request as described in Theys et al., 2021. OMI HCHO and $NO_2$ datasets are openly available on http://www.qa4ecv.eu/ecvs. TROPOMI Glyoxal data is available upon request as a part of the ESA S5p+I GLYRETRO project as detailed on the project website: https://glyretro.aeronomie.be/.

**Author Contributions**

PFL conceptualized, initiated, and managed this manuscript with contributions from IA, MB, TB, IDS, HE, CL, TS, DSZ, NT, MVR, PV, and TV. Formal analysis was carried out by MB, TB, IDS, HE, CL, and NT. DL and FR provided data curation and software support for TROPOMI HCHO data products. DSZ prepared, edited, and co-managed the manuscript with contributions from IA, MB, TB, IDS, HE, CL, PFL, TS, NT, MVR, PV, and TV.

**Competing Interests**

The authors declare that they have no conflict of interest.

**Acknowledgements**

We acknowledge financial support from the following projects: ESA S5P MPC (4000117151/16/I-LG); Netherlands Space Office TROPOMI Science Project; ESA S5p+Innovation GLYRETRO and ICOVAC projects (No. 4000127610/19/I-NS); Belgium Prodex TRACE-S5P (PEA 4000105598), and TROVA-2 (PEA 4000130630); Belgium BRAIN-2.be LEGO-BEL-AQ; EU FP7 QA4ECV project (grant no. 607405). This paper contains modified Copernicus data (2018/2020) processed by KNMI, BIRA-IASB, DLR, and SRON.

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
