# Peer review of "Air quality impacts of COVID-19 lockdown measures detected"

_Atmospheric Chemistry and Physics, 2021_

## Author Comment (AC2)

**RC2**: 'Review of acp-2021-534', Anonymous Referee #2, 20 Aug 2021  reply
In this study, results from more than 2 years of TROPOMI measurements of several tropospheric trace gases are reported and evaluated for possible impacts of the measures taken to reduce the spread of the Coronavirus in spring 2020. The approach taken is to compare data from 2020 to those from 2019 and, in the case of HCHO and CHOCHO, also to an OMI based climatology. In addition to the well-known reductions in NO2, decreases of different magnitudes are also found for SO2, HCHO, CHOCHO and CO over China and India.

The strong point of this manuscript is the combination of data on five trace gases, all measured from the same platform which tell a consistent story. These new data and comparisons are interesting and very relevant, are within the scope of ACP and should be published.

**General points**

1) The discussion of the reductions in NO2 columns on the other hand is neither new nor particularly interesting and I strongly suggest removing section 3, which displays data, which has already been shown in very similar ways in various previous publications. I simply do not see the benefit of repeating them here. Several published studies have gone beyond a simple comparison of data from 2020 and 2019 and have attempted corrections for meteorology, long-term trends and sampling, all of which are only discussed qualitatively here.

   **Response:**  We agree that Section 3 can be shortened with most of the discussion removed or moved to the Appendix, however we believe it should not be completely removed for several reasons. We want to give the reader context for the regions described in Sections 4 & 5. Secondly section 3, illustrates the consistent view TROPOMI offers from the global scale down to city-level (to our knowledge the multi-city panel figure 2 is a unique figure and of value to the reader). Lastly, section 3 also highlights our methodology: that is, first we analyzed the observed spatial and temporal patterns in TROPOMI NO2 data which further lead to the identification of regions of interest for the other four trace gases we present. Instead of discussion style, this shortened context-setting section will serve as a compact review of work that has been already published.

2) In the abstract and introduction, it is stated that this manuscript aims to "provide guidance on how to best interpret TROPOMI trace gas retrievals and to highlight how TROPOMI trace gas data can be used to understand event-based impacts on air quality from regional to city-scales around the globe". I do not think that the manuscript is doing this, and am not convinced that an ACP paper would be the right place to provide such guidance. I would suggest removing these statements throughout the paper and focusing on the science question.

   **Response:**  While the text stating the aims of our paper can be re-evaluated and modified as needed, we disagree that these statements should be removed completely. User guidance is critical for an ever-widening community of data users, especially for under-represented regions and user types. Beyond experienced

scientific users information needs to be made available so that (TROPOMI) data are properly utilized, analyzed, and interpreted. We maintain that proper user guidance is essential to ensure scientific integrity in an age of open data.

3) In the manuscript, the term "concentration" is often used where columns are meant. I think that these two quantities are not the same and would suggest that the authors search the manuscript for the term and replace it wherever they discuss columns. Also, the tropospheric columns retrieved from TROPOMI are not "averaged concentrations", nor "column concentrations", nor "column averaged amounts" but integrated concentrations. Please correct.

**Response:** Appearances of term 'concentration' in general discussion portions of the text will not be modified. We have, however, checked for all appearance of the term 'concentration(s)' when in direct reference to TROPOMI column measurements and for clarity these appearances as listed below will be replaced with the term 'column amount' where appropriate.

- Line 18-19: We report clear COVID-19-related decreases in NO2 concentrations on all continents."
- Line 59, 64: Indirectly referring to TROPOMI observed concentrations
- Lines 99-100: "Using the spectral radiance measurements from TROPOMI, atmospheric concentrations of different gases are retrieved as well as cloud and aerosol properties."
- Line 201-202: "First we compare the concentrations in 2020 with those from the same period..."
- Line 324-325: "For this paper, we correct the HCHO concentrations for this meteorological impact prior to using the data in the analyses."
- Line 378-379: " A strong reduction in the NO2 tropospheric concentration of about 40% is observed over Santiago during this period"
- Line 474: "Figure 3 and Figure 4 illustrate the tropospheric concentration of NO2 over Europe,..."
- Line 541-542: "This is clearly illustrated in the upper panel of Figure 5 showing CHOCHO concentrations,"
- Line 568: Caption Figure 6, (correct to 'median tropospheric column amount)
- Line 616-617: "The small delay between the initial decrease in $NO_2$ concentration and the observed decreases in the other trace gas signals"
- Line 631-632: "$NO_2$ and $SO_2$ the concentrations are clearly lower across the country in 2020 as compared to 2019. Although less prominent, concentrations of CO, HCHO, and CHOCHO appear to be lower in April 2020"
- Line 638: Caption Figure 7
- Line 647, 649: References to Fig 7 and Fig 8a with "NO2 column concentrations"
- Line 665-666: Caption Figure 8
- Line 722-723: "HCHO column concentrations"
- Line 766-769: TROPOMI observed "NO2 concentrations"
- Line 792: "CHOCHO concentrations"
-

We also propose to modify two sentences (lines 189-192) as follows, "TROPOMI observes atmospheric concentrations of trace gases integrated over a vertical column, which is not the same as a direct measurement of the (near-surface) emission. The amount of a given trace gas integrated over a vertical column at a certain location depends not only on emission and deposition, but also on atmospheric transport and (photo)chemical reactions."

4) The strength of this paper is the combination of results from several trace gases in a consistent way. I, therefore, would suggest that also the units used for the different columns are consistent between molecules and between different figures and tables. I do not see any advantage in using different units for NO2 than for HCHO, CHOCHO and CO other than that these are the units provided in the operational product. Please make consistent.

   **Response:** The units in the revised manuscript for all figures and when mentioned in the text will be in the TROPOMI standard base unit of mol/m2, making note that for figure and color bar label readability both millimol/m2 and micromol/m2 are sometimes used. Additional, supporting information and explanation of the TROPOMI data file units can also be provided in Appendix A.

5) In some places, the article reads like a TROPOMI advertisement. I do not think this is necessary – the great TROPOMI data and figures presented speak for themselves and I do not see the need to highlight the "societal relevance of the TROPOMI mission" in a scientific paper.

   **Response:** We have evaluated some of the 'claim' statements and will modify several in the revised manuscript. We, however tend to disagree on the point of societal relevance as TROPOMI data is increasingly utilized to address a variety of societally relevant issues.

**Specific points**

1) Line 98: "absorption regions for clouds" – this sounds a bit odd. I assume that the absorption bands of O2 are referred to which are used for cloud products.
   **Response:** will be reworded for clarity in revised manuscript, Suggestion: spectral bands selected to measure the absorption by a large number of trace atmospheric constituents as well by clouds and aerosols.

2) Line 112 – 120: I would suggest removing this paragraph
   **Response:** These sentences were added to briefly illustrate and explain why we go beyond qualitative comparison. We do agree to streamline text as needed using comparable language throughout for specific mentions of methodologies employed (per species) to make our comparisons more quantitative.

3) Table 1: CHOCHO: Not sure, why Lerot 2010 is cited here and why the comment on precursors made for HCHO is not repeated for CHOCHO
   **Response:**  will be updated in revised manuscript

4) Line 203 – 204: Sentence appears to be incomplete
   **Response:** will be reworded in revised manuscript

5) Line 338: operations => operational
   **Response:** will be corrected in revised manuscript

6) Line 532: sentence appears to be incomplete.

**Response:** Yes, can be reworded for revised manuscript; Suggestion:  City-scale impacts of lockdown on NO2 tropospheric column amounts for Wuhan and Beijing _are presented_ in Sect. 3.

7) Figure 5: I would suggest adding difference plots as a third line.
**Response:** Given the different lifetimes of species, meteorological differences, among other reasons described in Section 2, a simple difference plot will include many more features of trace gas variability than just the reduction of emissions due to COVID-19 lockdown measures, and is likely to be more confusing to the reader and may add little value.

8) Figure 6: Something is not right with the glyoxal columns – they extend into the future!
**Response:** This will be corrected, and as described in our response to Reviewer #1, given the data availability, all time series will be extended through Nov 2020.

9) Figure 6 and elsewhere: The region over which the data is averaged is called Northern China. However, this region is not particularly far to the North of China but rather in the central East of the country.
**Response:** this will be clarified in the revised manuscript

10) Line 617: I am not convinced by the explanation given for the delay in reductions in VOC and SO2 compared to NO2. To my knowledge, the difference in lifetimes is of the order of hours and secondary HCHO formation is a matter of hours or maybe days but not weeks.
**Response:** this will be clarified in the revised manuscript, however recent literature (for which we will add an explicit citation) supports the stated lifetime.

11) Figure 7: I would suggest adding difference plots as a third line.
**Response:** See response regarding same comment for item 7) regarding Figure 5.

12) Line 794: Sentence appears to be incomplete
Response: will be updated in revised manuscript.  Suggestion, "For HCHO, after correcting for the effect of seasonal and temperature variations, we observe a coincident 40% _reduction_."

13) Figure C2: While I am convinced that the COBRA SO2 product is better than the DOAS SO2-product, this figure does not prove that. The figure mainly shows the difference in absolute values of the two products.
**Response:** will be updated as needed for clarity

14) Figure D2: I did not know what an "ombrothermic diagram" is before checking so maybe other readers would also benefit from an explanation here. I also think that the caption is not correct as the two columns are for different regions and not for temperature and precipitation as stated.
**Response:** will be updated as needed for clarity

---

## Author Response (AR1)

**Consolidated, Point-by-Point Review Response Document**

**Response and Revision for Review #1 Comments**

**RC1**: 'Comment on acp-2021-534', Anonymous Referee #1, 08 Aug 2021  reply
This paper presents the observed changes in the atmospheric column amounts of five trace gases ($NO_2$, $SO_2$, CO, HCHO and CHOCHO) detected by TROPOMI to investigate the reductions of anthropogenic emissions due to COVID-19 lockdown measures in 2020. It aims to provide guidance to data users on how to best interpret and analyze TROPOMI trace gas data not only for lockdown-driven emission changes but also for other event-driven changes. I would suggest small revisions before the publication.

General comments (Reviewer #1):

1. The authors used different quality assurance values (qa_value) for different species. Are the definition of qa_value consistent among species? If not, a small summary is appreciated. Otherwise, it is very confusing for readers.

   **Response:** As a general guideline, application of a qa_value greater than 0.5 ensures a basic level of quality assurance for all operational TROPOMI data products. However, some data products have provided additional qa_value thresholds relevant for certain scenarios as described in the respective TROPOMI Product Readme File (PRF) documents and as listed in Appendix A. For this paper, only $NO_2$ differs, with the application of qa_value of greater than 0.75, and this is described in the second to last paragraph in Section 2.2 (lines 234-238). A qa_value of greater than 0.5 was applied the other operational TROPOMI data products used in this paper (CO and HCHO) as described in Sections 2.4 and 2.5, respectively.

   To aid the reader we will update the text and listing in Appendix A to make it clear which qa_value threshold was applied to each of the operational TROPOMI data products. We will also make it clear in the text and Table 1 which data products are operational and which are prototypes.
   **Revision:**
   A descriptor has been added to the first column of Table 1 to describe if a data product is an operational or prototype which is consistent with product-specific subsections in Section 2.
   The entry in the PRF row in the last column of Table A1 in Appendix A has been updated to include the recommended qa_values as described in the product-specific subsections in Section 2.

2. The authors used different approaches to consider the contributions from natural sources and meteorology for different species. I would appreciate a table or graph to summary the approaches.

   **Response:** We think that a table would be insufficient to clarify how the seasonal, meteorological, temperature-driven, and other effects have been accounted for in the analysis of the data products presented in this paper. These methods are currently described per species in Sections 2.2 through 2.6. We will however ensure in the revised manuscript that it is clear to the reader how the treatments differ per species as described in Section 2.

**Revision:** In response to specific review points, minor changes have been applied to Section 2.2 through 2.6 which, in general improve overall clarity to section 2. Additional information is available in Appendix A for example, summarizing which qa_values were used.

Specific comments (Reviewer #1)::

1. Table 1. Are power plants primary sources of CO? Please confirm.

   **Response:** To answer this question we first clarify what is meant by the word 'primary'. In the sense of primary or foremost, carbon monoxide (CO) emitted from power plants is not the largest source of CO. Based on the EDGAR v5.0 database which gives 2015 emissions per sector on a global basis, CO from the Power industry contributes just over 1%. The largest CO sources are transportation, residential cooking and heating, other industrial combustion (ex. Iron and steel production industry), and agricultural burning (which is incorporated in the broader label of biomass burning used in our paper).

   In the sense of direct/primary emission, power generation is a primary source of CO as it is emitted directly from power plants.

   As a result of this question, we will modify the table header label 'Primary emission sources' and update the order of appearance of the sources so that it is more apparent to the reader which sources are the largest. It should be noted, however, that based on analysis from the previous EDGAR emissions database (2010) the relative importance of largest sources of CO can vary per region and/or country (Janssens-Maenhout et al., 2015).
   **Revision:**
   To clarify what is meant by 'primary' emissions we have changed the caption of Table 1 and the header descriptor in the last column of Table 1 to 'Main Emission Sources' to indicate to the reader that we are providing description of the largest emission sources. We have also changed the order of the listing of the CO emission sources in Table 1 so that the largest are listed first.
   In support, a sentence is added to second paragraph of section 2, "However, it is noted that the relative contribution of these sources varies per global region (Janssens-Maenhout et al., 2015)."

2. Line 187. "In future studies, the averaging kernels could be used for inversion modelling of emissions thus eliminating this error completely." It is not clear to me how the error can be eliminated completely. Please clarify.

   **Response:** To better explain how the profile shape related error is eliminated, the sentence cited above is revised as follows,
   **Revision:**
   "In future studies, the averaging kernels could be used for inversion modelling of emissions. As explained in Eskes and Boersma 2003, relative comparisons between the observations and the model used in the inverse modelling system, and therefore the resulting emissions, no longer depend on the retrieval a-priori profile shape when the kernel is applied to the model."

3. Line 204. Please clarify what are changes driven by mechanisms.

   **Response:**  This sentence is updated for improved clarity.
   **Revision:**  "First we compare the concentrations in 2020 with those from the same period from earlier years and then carry out additional analysis to compare the lockdown-driven variability with seasonal and meteorological variability taking in account local information about lockdown and anticipated impacts from different source sectors."

4. Line 230. Please add reference for the magnitude of 20-60%.

   **Response, Revision:**  We have added a reference in the revised manuscript to support this statement with the NO2 ATBD (van Geffen et al., 2021).

5. Line 265. Please add reference for the error.

   **Response:** The reference Theys et al., 2021 has been added to this sentence in the revised manuscript.

6. Line 327. What is the term of novel applied to? The algorithm?

   **Response:** the term 'novel' is applied to the temperature correction for HCHO. We propose the following
   **Revision:** "This temperature correction is performed for each region and on the OMI and TROPOMI time series separately".

7. Line 338. Operational products instead of operations products.

   **Response:** This sentence will be corrected in the revised manuscript.
   **Revision:** operations is corrected to 'operational'

8. Line 346. Please define "box-air mass factors" before use.

   **Response:** To address this point we propose to modify the text and references as follows,
   **Revision:** "Air mass factors are calculated following the formulation of Palmer et al. (2001), which combines altitude-dependent air mass factors (or Box-AMFs) with a priori glyoxal concentration profiles. The Box-AMFs represent the instrumental sensitivity to changes in concentration at any altitude and are precomputed using the radiative transfer model VLIDORT v2.7 (Spurr and Christi, 2019), while the a priori profiles are provided by the MAGRITTE chemistry-transport model (Müller et al., 2018, 2019)."

9. Figure 6 & 9. The data for June-Dec, 2020 is already available. Is there any specific reason to exclude them from the figures?

   **Response:**  At the time of manuscript preparation, the cutoff date was in part limited by the availability of the ERA5 surface temperature data needed for HCHO temperature correction. Now that this is no longer the case, the time series presented in these plots will be extended through end-November 2020. A data processing update carried out in early December 2020 affects the tropospheric column amounts and causes a discontinuity in the NO2 time series.

This discontinuity will only be resolved once the operational reprocessed dataset is available (planned for the second half of 2022). As such, December 2020 will not be included.
**Revision:** Figures 6 and 9 have been updated to extend the time series for 2020. Corresponding figure captions. For consistency, all other time series plots (Figures 2, 3, 5, and 8) have been similarly extended.

10. Line 746. Is there any other evidences/reports from literatures to support the explanation for the enhanced CO?

**Response:** We have examined this issue and can, based on references, state generally that fires play a role in the observed, enhanced CO. However, to more fully answer the question of why CO in 2020 is higher than 2019, is only possible by employing chemical transport model calculations and this is beyond the scope of this work.
**Revision:** We updated the text and references as follows:

Original text: "Figure 7 shows that the CO amounts in southern India are higher in 2020 compared to 2019. The reason could be the accumulation of CO originating from elsewhere prior to the lockdown period."

Revised text: "Figure 7 shows that the CO amounts in southern India are higher in 2020 as compared to 2019. The enhanced CO values in 2019 and 2020 are detected above regions (e.g. Madhya Pradesh, Odisha, and Chhattisgarh) where seasonal forest fires commonly occur in April/May (Chandra and Kumar Bhardwaj, 2015, Srikanta et al. 2020). Thus, the enhancement of CO for the different years depends not only on the fire activity but also on how the meteorological situation prevents or permits the accumulation of CO in the atmosphere. To more fully address the reasons why CO is higher in 2020 than 2019, future studies could carry out calculations using a chemical transport model."

The following references have been added:
Chandra, K. K., and Kumar Bhardwaj, A.: Incidence of forest fire in India and its effect on terrestrial ecosystem dynamics, nutrient and microbial status of soil, International Journal of Agriculture and Forestry, 5(2), 69-78, doi:10.5923/j.ijaf.20150502.01, 2015.

Srikanta, S., Pilla, F., Basu, B., Sarkar Basu, A., Sarkar, K., Chakraborti, S., Kumar Joshi, P., Zhang, Q., Wang, Y., Bhatt, S., Bhatt, A., Jha, S., Keesstra, S., and Roy, P. S.: Examining the effects of forest fire on terrestrial carbon emission and ecosystem production in India using remote sensing approaches, Sci. Tot. Environ., 725, doi:10.1016/j.scitotenv.2020.138331, 2020.

11. Figure D2. The font size is too small to read.

**Response, Revision:** This font size has been increased to improve readability in the revised manuscript. The caption has also been updated accordingly.

**Response and Revision for Review #2 Comments**

**RC2**: 'Review of acp-2021-534', Anonymous Referee #2, 20 Aug 2021  reply
In this study, results from more than 2 years of TROPOMI measurements of several tropospheric trace gases are reported and evaluated for possible impacts of the measures taken to reduce the spread of the Coronavirus in spring 2020. The approach taken is to compare data from 2020 to those from 2019 and, in the case of HCHO and CHOCHO, also to an OMI based climatology. In addition to the well-known reductions in NO2, decreases of different magnitudes are also found for SO2, HCHO, CHOCHO and CO over China and India.

The strong point of this manuscript is the combination of data on five trace gases, all measured from the same platform which tell a consistent story. These new data and comparisons are interesting and very relevant, are within the scope of ACP and should be published.

**General points**

1) The discussion of the reductions in NO2 columns on the other hand is neither new nor particularly interesting and I strongly suggest removing section 3, which displays data, which has already been shown in very similar ways in various previous publications. I simply do not see the benefit of repeating them here. Several published studies have gone beyond a simple comparison of data from 2020 and 2019 and have attempted corrections for meteorology, long-term trends and sampling, all of which are only discussed qualitatively here.

   **Response:**  We agree that Section 3 can be shortened with most of the discussion removed or moved to the Appendix, however we believe it should not be completely removed for several reasons. We want to give the reader context for the regions described in Sections 4 & 5. Secondly section 3, illustrates the consistent view TROPOMI offers from the global scale down to city-level (to our knowledge the multi-city panel figure 2 is a unique figure and of value to the reader). Lastly, section 3 also highlights our methodology: that is, first we analyzed the observed spatial and temporal patterns in TROPOMI NO2 data which further lead to the identification of regions of interest for the other four trace gases we present. Instead of discussion style, this shortened context-setting section will serve as a compact review of work that has been already published.
   **Revision:** Several key references to works describing TROPOMI capabilities as well as global to city-based COVID-19 lockdown-driven changes in emissions have been cited. Large portions of the discussion-style text from Section 3 have been moved to Appendix B in support of the detailed lockdown timing overview table and set of references. The first objective (original line 66) has been revised as well as the description of Section 3 in the last paragraph of Section 1.

2) In the abstract and introduction, it is stated that this manuscript aims to "provide guidance on how to best interpret TROPOMI trace gas retrievals and to highlight how TROPOMI trace gas data can be used to understand event-based impacts on air quality from regional to city-scales around the globe". I do not think that the manuscript is doing this, and am not convinced that an ACP paper would be the right place to provide such guidance. I would suggest removing these statements throughout the paper and focusing on the science question.

**Response:** While the text stating the aims of our paper can be re-evaluated and modified as needed, we disagree that these statements should be removed completely. User guidance is critical for an ever-widening community of data users, especially for under-represented regions and user types. Beyond experienced scientific users information needs to be made available so that (TROPOMI) data are properly utilized, analyzed, and interpreted. We maintain that proper user guidance is essential to ensure scientific integrity in an age of open data.

**Revision:** The first sentence in the abstract has been modified as follows, "The aim of this paper is to highlight how TROPOMI trace gas data can best be used and interpreted to understand event-based impacts on air quality from regional to city-scales around the globe."

3) In the manuscript, the term "concentration" is often used where columns are meant. I think that these two quantities are not the same and would suggest that the authors search the manuscript for the term and replace it wherever they discuss columns. Also, the tropospheric columns retrieved from TROPOMI are not "averaged concentrations", nor "column concentrations", nor "column averaged amounts" but integrated concentrations. Please correct.

**Response, Revision:** Appearances of term 'concentration' in general discussion portions of the text will not be modified. We have, however, checked for all appearance of the term 'concentration(s)' when in direct reference to TROPOMI column measurements and for clarity these appearances as listed below will be replaced with the term 'column amount' where appropriate.

- Line 18-19: We report clear COVID-19-related decreases in $NO_2$ concentrations on all continents."
- Line 59, 64: Indirectly referring to TROPOMI observed concentrations
- Lines 99-100: "Using the spectral radiance measurements from TROPOMI, atmospheric concentrations of different gases are retrieved as well as cloud and aerosol properties."
- Line 201-202: "First we compare the concentrations in 2020 with those from the same period…"
- Line 324-325: "For this paper, we correct the HCHO concentrations for this meteorological impact prior to using the data in the analyses."
- Line 378-379: " A strong reduction in the $NO_2$ tropospheric concentration of about 40% is observed over Santiago during this period"
- Line 474: "Figure 3 and Figure 4 illustrate the tropospheric concentration of $NO_2$ over Europe,..."
- Line 541-542: "This is clearly illustrated in the upper panel of Figure 5 showing CHOCHO concentrations,"
- Line 568: Caption Figure 6, (correct to 'median tropospheric column amount)
- Line 616-617: "The small delay between the initial decrease in $NO_2$ concentration and the observed decreases in the other trace gas signals"
- Line 631-632: "$NO_2$ and $SO_2$ the concentrations are clearly lower across the country in 2020 as compared to 2019. Although less prominent, concentrations of CO, HCHO, and CHOCHO appear to be lower in April 2020"
- Line 638: Caption Figure 7
- Line 647, 649: References to Fig 7 and Fig 8a with "$NO_2$ column concentrations"
- Line 665-666: Caption Figure 8
- Line 722-723: "HCHO column concentrations"
- Line 766-769: TROPOMI observed "$NO_2$ concentrations"
- Line 792: "CHOCHO concentrations"

We have also modified two general sentences (original lines 189-192) as follows, "TROPOMI observes atmospheric concentrations of trace gases integrated over a vertical column, which is not the same as a direct measurement of the (near-surface) emission. The amount of a given trace gas integrated over a vertical column at a certain location depends not only on emission and deposition, but also on atmospheric transport and (photo)chemical reactions."

4) The strength of this paper is the combination of results from several trace gases in a consistent way. I, therefore, would suggest that also the units used for the different columns are consistent between molecules and between different figures and tables. I do not see any advantage in using different units for NO2 than for HCHO, CHOCHO and CO other than that these are the units provided in the operational product. Please make consistent.

**Response:** The units in the revised manuscript for all figures and when mentioned in the text will be in the TROPOMI standard base unit of mol m$^{-2}$, making note that for figure and color bar label readability both millimole m$^{-2}$ and micromole m$^{-2}$ are sometimes used. Additional, supporting information and explanation of the TROPOMI data file units can also be provided in Appendix A.
**Revision:** Figures 2 through 9 have been updated to depict TROPOMI units of mol m$^{-2}$. Figure captions have been updated accordingly. Reference to the product-specific Product User Manual (PUM) documents in Appendix A provides a link to additional detail about TROPOMI data formatting including units.

5) In some places, the article reads like a TROPOMI advertisement. I do not think this is necessary – the great TROPOMI data and figures presented speak for themselves and I do not see the need to highlight the "societal relevance of the TROPOMI mission" in a scientific paper.

**Response:** We have evaluated some of the 'claim' statements and will modify several in the revised manuscript. We, however tend to disagree on the point of societal relevance as TROPOMI data is increasingly utilized to address a variety of societally relevant issues.
**Revision:** Two sentences (original text lines 52, 79) have been modified in the revised text.

**Specific points**

1) Line 98: "absorption regions for clouds" – this sounds a bit odd. I assume that the absorption bands of O2 are referred to which are used for cloud products.
**Response, Revision:** has been reworded for clarity in revised manuscript as follows, "…spectral bands selected to measure the absorption by a large number of trace atmospheric constituents as well as by clouds and aerosols."

2) Line 112 – 120: I would suggest removing this paragraph
**Response:** These sentences were added to briefly illustrate and explain why we go beyond qualitative comparison. We do agree to streamline text as needed using comparable language throughout for specific mentions of methodologies employed (per species) to make our comparisons more quantitative.
**Revision:** In response to specific review points, minor changes have been applied to improve overall clarity to section 2 describing methodologies.

Additional information is available in Appendix A for example, summarizing which qa_values were used.

3) Table 1: CHOCHO: Not sure, why Lerot 2010 is cited here and why the comment on precursors made for HCHO is not repeated for CHOCHO
**Response, Revision:** Table 1 has been updated in revised manuscript where, 2010 has been removed and HCHO and CHOCHO source and lifetime entries have been harmonized.

4) Line 203 – 204: Sentence appears to be incomplete
**Response:** This sentence is updated for improved clarity.
**Revision:** "First we compare the concentrations in 2020 with those from the same period from earlier years and then carry out additional analysis to compare the lockdown-driven variability with seasonal and meteorological variability taking in account local information about lockdown and anticipated impacts from different source sectors."

5) Line 338: operations => operational
**Response, Revision:** this has been corrected in the revised manuscript.

6) Line 532: sentence appears to be incomplete.
**Response, Revision:** Yes, this sentence has been reworded in the revised manuscript as follows, "City-scale impacts of lockdown on NO2 tropospheric column amounts for Wuhan and Beijing _are presented_ in Sect. 3."

7) Figure 5: I would suggest adding difference plots as a third line.
**Response:** Given the different lifetimes of species, meteorological differences, among other reasons described in Section 2, a simple difference plot will include many more features of trace gas variability than just the reduction of emissions due to COVID-19 lockdown measures, and is likely to be more confusing to the reader and may add little value. Therefore we will not add this plot.

8) Figure 6: Something is not right with the glyoxal columns – they extend into the future!
**Response, Revision:** Figure 6 has been corrected, and as described in our response to Reviewer #1, given the data availability, each time series for 2020 shown in this paper has been extended until 1 December 2020.

9) Figure 6 and elsewhere: The region over which the data is averaged is called Northern China. However, this region is not particularly far to the North of China but rather in the central East of the country.
**Response:** we have investigated the demarcation of regions relevant to Chinese air quality (as described, for example in Song et al., 2017, https://doi.org/10.1016/j.envpol.2017.04.075) and we agree that while our defined region is not exclusively Northern China as does include some of eastern and central China, the predominant area within the box is Northern China (not to be confused with northeastern China). Thus, for the sake of simplicity we will continue to refer to this region as northern China.

10) Line 617: I am not convinced by the explanation given for the delay in reductions in VOC and SO2 compared to NO2. To my knowledge, the difference in lifetimes

is of the order of hours and secondary HCHO formation is a matter of hours or maybe days but not weeks.

**Response, Revision:** the revised sentence now includes a reference (Stavrakou et al., 2021) supporting the role of secondary HCHO formation and how it differs from NO2.

11) Figure 7: I would suggest adding difference plots as a third line.
**Response:** See our response regarding same comment for item 7) regarding Figure 5.

12) Line 794: Sentence appears to be incomplete
**Response, Revision:** This sentence has been updated in revised manuscript as follows, "For HCHO, after correcting for the effect of seasonal and temperature variations, we observe a coincident 40% _reduction_."

13) Figure C2: While I am convinced that the COBRA SO2 product is better than the DOAS SO2-product, this figure does not prove that. The figure mainly shows the difference in absolute values of the two products.
**Response:** will be updated as needed for clarity
**Revision:** the caption has been revised and includes reference to Theys et al. 2021.

14) Figure D2: I did not know what an "ombrothermic diagram" is before checking so maybe other readers would also benefit from an explanation here. I also think that the caption is not correct as the two columns are for different regions and not for temperature and precipitation as stated.
**Response:** will be updated as needed for clarity.
**Revision:** The figure caption has been updated to more clearly describe the temperature and precipitation information depicted for the two regions shown.